# Financial Leverage and Debt Maturity Targeting: International Evidence

**Ali Gungoraydinoglu and Özde Öztekin \***

College of Business, Florida International University, 11200 S.W. 8th Str., Miami, FL 33199, USA; agungora@fiu.edu

\* Correspondence: ooztekin@fiu.edu

**Abstract:** We provide evidence on leverage and debt maturity targeting in a large international setting. There are key differences in the relative importance of institutional factors in explaining actual as opposed to target capital structures. Targets and target deviations are plausibly influenced by the institutional environment. Firms from countries with strong legal institutions target lower leverage and higher long-term debt, whereas better-functioning financial systems result in lower target leverage and long-term debt. Financial crisis has shifted the desired structure of the securities toward shorter maturities and has led to more prevalent target deviations. Better institutions significantly decrease the likelihood of target deviations.

**Keywords:** target capital structure; debt maturity; international; institutions

## 1. Introduction

The importance of capital structure for corporate and real outcomes has been extensively recognized in the extant literature for both small- and medium-sized enterprises (SMEs) and listed firms from developed as well as developing economies (e.g., Anton 2019). An influential view in the theory of capital structure is the existence of an optimal capital structure (Graham and Harvey 2001; DeAngelo et al. 2011; Frank and Goyal 2015; DeMarzo and He 2021). Accordingly, a firm's financing choice is not only reflected by its observed debt ratio but also by its target debt ratio and its deviation from the target. Such a target can be established by trading off the costs and benefits of debt, including bankruptcy costs, tax benefits, and agency costs. The credibility of this capital structure model hinges on the nature of the target determination and the adjustment process. If the estimates of target capital structures are meaningful, the cross-sectional tests of the fitted targets and target deviations should reliably reflect these costs and benefits.

Two main empirical predictions related to leverage targeting behavior have been confirmed in both U.S. and international samples: the convergence of the actual debt ratio toward a target or an optimum in the long run and consistent cross-sectional relationships between the observed debt ratios and several firm, industry, macroeconomic, and national characteristics, such as profitability, tax status, asset type, and quality of institutions. However, no research has yet directly tested and established whether the estimates of target capital structures are meaningful. This article addresses this research gap by investigating the significance and the determinants of capital structure targets and target deviations and systematically testing the estimated leverage and debt maturity targets and target deviations across different environments.

Many studies have evaluated whether firm actions inferred from various leverage models are consistent with the predictions of the theory. The studies testing the tradeoff theory of capital structure typically regress observed leverage ratios on the proxies for the

costs of financial distress and agency conflicts and tax benefits of debt with the idea that the model is a good first order approximation of the equilibrium. This approach is problematic since firms do not typically operate at their optimal leverage due to financial frictions, indicating that the observed capital structure is a noisy proxy of target capital structure. In contrast to this traditional approach, this paper proposes a new empirical test of target behavior based on capital structure targets and target deviations. If the estimated capital structure targets are relevant, the influence of factors such as bankruptcy costs, agency costs, and information asymmetry costs on firms' target capital structures should be congruent with the theory. Furthermore, a firm that is not at the leverage target should take actions that adjust toward the target to close its deviation from the target. These actions should differ across firms depending on their environment, as different conditions require firms to react to shocks differently. The theory would be reliable to the extent that it correctly predicts the determination of the capital structure targets and target deviations based on the institutional, financial, and macroeconomic environment in which the firm operates. Thus, while the current debate about the role of national institutions is focused around observed leverage and observed debt maturity as a proxy for optimal leverage and optimal debt maturity, this study estimates optimal leverage and optimal debt maturity with various reliable empirical models and uses these estimates as its key measures in subsequent tests.

The proposed approach for examining the cross-sectional determinants of target capital structures and target deviations is an improvement over previous research examining the cross-sectional determinants of observed leverage ratios for three important reasons. First, if firms cannot instantaneously offset the movements from their optimal capital structure, as posited by the dynamic tradeoff theory, cross-sectional variations in actual capital structures could be observed, even across a sample of firms sharing the same target capital structures. Second, shocks to the profitability and investment opportunities of firms, stock prices, and other factors may cause variations in target debt ratios over time without necessarily having the same impact on the observed debt ratios. Third, the extent to which various cost and benefit factors affect the observed leverage ratios as opposed to the leverage targets could vary across firms and/or over time. Consequently, leverage models that assume that firms always operate at their long-run equilibrium and proxy for target capital structures using observed capital structures while also assuming that the target deviations are statistically and economically trivial fail to properly account for the true firm capital structure dynamics.

To understand this motivation, it is useful to illustrate the importance of country-level factors for the observed leverage relative to the target leverage. Variance decomposition analyses reveal that the ordering of the importance of the institutional factors differs sharply for actual leverage as opposed to target leverage in the overall sample. Among the twelve selected institutional factors, legal origin ranks ninth by capturing only 3% of the total variation in the observed leverage but comes first by explaining 36% of the total variation in target leverage. On the other hand, internal and external governance are the two least important institutional features for explaining the total variation in target leverage but are among the top three (ranking right below the taxes) for explaining the total variation in the observed leverage. These results suggest that misleading conclusions may be reached regarding the relative influences of institutional factors for the leverage targeting behavior of corporate firms since observed leverage is a noisy proxy of target leverage.

This article contributes to this literature by systematically testing estimated leverage targets and target deviations across different institutional, financial, and macroeconomic environments. Motivated by the above discussion, we evaluate four distinct hypotheses. First, the extent to which various cost and benefit factors affect observed leverage ratios as opposed to leverage targets and/or target deviations that differ across firms and/or over time. Second, estimated targets are meaningfully and significantly different across institutional environments, reflecting the various costs and benefits of the financing imposed on firms. Specifically, in environments that are more conducive to faster capital structure

adjustments, ceteris paribus, firms would be closer to their leverage target. Third, the differences in the targeting behavior of firms in weak and strong institutional environments are expected to be more pronounced during a financial crisis when the external financing constraints are more likely to be more binding. Finally, all else being equal, it is expected that persistent deviations from the target would be less likely in strong institutional settings and during better times, leading to lower financial frictions, preventing firms from undertaking leverage adjustments. Consequently, financial crises are associated with a greater likelihood of target deviations, particularly for the financially constrained firms that face higher external financing costs.

To date, no research has assessed the relevance of the country-level institutional factors for the choices of leverage and debt maturity targets and target deviations in a dynamic and global setting. This study closes this research gap and makes several important contributions to the literature. First, while many papers assess the determinants of a firm's capital structure, most of these studies examine firms within a single country, typically the United States. Studies with an international scope typically ignore the dynamic nature of the data or focus on observed (and not target) capital structure in cross-sectional tests. For example, Fan et al. (2012) examine how the institutional environment influences observed capital structure choices. Their analysis implicitly assumes the instantaneous adjustment to optimal capital structure and mostly emphasizes supply side financing (i.e., investors). While the empirical analysis is dynamic in some papers, it does not assess institutional influences on debt maturity choices, and it does not focus on targets. In addition, in contrast to some other studies (e.g., Fan et al. 2012), this study employs a large set of institutional features that are not only related to the supply side but that are also related to the demand side (i.e., corporations), reflecting various costs (e.g., bankruptcy costs, agency costs, and information asymmetry costs) that firms face in their country. Second, because any test of optimal capital management is a joint test of capital structure and the efficiency of the empirical model adopted to estimate the targets, empirical analysis is conducted using a variety of established target estimation techniques to ensure that a particular model specification does not drive the results. Specifically, target capital structure is estimated using three distinct empirical approaches: Blundell and Bond's (1998) approach, Fama and French's (2002) approach, and a simple median industry debt ratio. Third, the international sample is the most comprehensive of the capital structure studies conducted to date and spans 46 countries and 29 years (compared to 37 countries and 16 years, excluding crisis times, in the extensive samples studied by Öztekin and Flannery 2012; Gungoraydinoglu and Öztekin 2011; Öztekin 2015). The large cross-section and longer time series not only provide tests that are more powerful but also yield novel results. The availability of post-2006 data creates a novel environment in which to demonstrate how the global financial crisis has affected target choices and target deviations. The recent global financial crisis had a significant impact on firms around the world, yet little is known about how it affected capital structure choices and the adjustment to target leverage of firms. This study adds to the existing literature by testing the impact of the recent financial crisis on leverage targets and target deviations for the first time. The current study also analyzes how the recent global financial crisis has affected the relationships among leverage targets and a country's institutional strength. Finally, the use of three distinct leverage measures: (1) absolute level of total borrowing; (2) absolute level of long-term borrowing, and (3) the mix of long-term debt in total debt—allow for more refined tests on the impact of the institutional, financial, and macroeconomic environment on the capital structure targeting behavior of firms.

Our empirical investigation yields several important, novel insights. Our first result is that there are important differences in the relative importance of institutional and financial development determinants in explaining the total variation in actual and target leverage, both in terms of their magnitudes as well as their rankings. These findings lend support to the idea that testing the cross-sectional determinants of target capital structures is distinct from examining the cross-sectional determinants of observed leverage ratios.

Our second result is that different environments impose different costs and benefits of financing on firms, and these are plausibly reflected in the estimated leverage targets. Firms from countries with a common law origin legal system, stronger shareholder protection, better internal and external governance, stronger bankruptcy procedures, more concentrated ownership and control, and lower information asymmetry costs on debt and equity target lower leverage and higher long-term debt, whereas a financial structure based on the effectiveness of capital markets (market-based financial system) rather than intermediaries (bank-based financial system), better-functioning financial systems (as classified by Levine 2002), lower tax benefits, and stronger creditor protection are associated with lower target leverage and long-term debt targets.

Our third result is that the financial crisis has affected the desired structure of the securities that are offered, shifting them toward shorter maturities. Depending on the target estimation methodology, on average, firms targeted up to 2.5 percentage points more total debt (24.29% vs. 26.78%) and 1.5 percentage points less long-term debt (11.20% vs. 9.75%) in their capital structures during the crisis period (2007–2008) compared to during the pre-crisis period (1989–2006), translating into relative changes of 10% and 13% for total and long-term debt targets, respectively. Furthermore, despite some exceptions, the differences in the targeting behavior of firms in weak and strong institutional environments are generally wider during the financial crisis compared to during normal times. Put differently, the adverse effects of market conditions on the financing patterns of firms are felt more strongly in countries with weaker legal and financial institutions that subject firms to higher distress costs, contracting costs, moral hazard problems, agency costs of debt and equity, and information asymmetry costs. Mixed results with some institutional variables are partly due to their lower explanatory power for leverage targets compared to observed leverage during crisis times.

Our fourth result is that the variation in external financing costs, as captured by the quality of legal institutions, the financial development of the country, and the financial crisis times, is an important driver of target deviations. Consistent with the theory, deviations from optimal leverage are less prevalent with better institutions and financial systems where the financial frictions are less binding. Furthermore, after controlling for industry and year fixed effects to account for any shocks to profitability, investment, and stock prices as well as a firm's initial deviation, the country-level indicators of stock market development, bond market development, and the availability of private credit are associated with a lower likelihood of target deviations. Specifically, the odds of deviating are lower by up to 17 percentage points with higher stock market capitalization, up to 22 percentage points with higher bond market capitalization, and up to 25 percentage points with a higher supply of private credit. Financial frictions are significantly higher in hard times. The likelihood of target deviations was up to three times higher for an average firm during the recent global financial crisis. This effect was more prevalent among the financially constrained firms that face relatively higher costs in accessing external capital markets, raising their probability of deviations up to five times.

This paper establishes the various costs and benefits of financing on firms using estimated leverage targets and target deviations in a large international setting. We contribute to the literature by modeling leverage targets and target deviations as a function of institutional, financial, and macroeconomic factors, testing the impact of the recent financial crisis on leverage targets and deviations, and analyzing how the recent financial crisis has affected the relationships between leverage targets and a country's institutional strength. Overall, estimates of target leverage and deviations are meaningful and congruent with theory, and choices of firms on optimal capital structure vary plausibly across institutional, financial, and macroeconomic environments around the globe. Hence, our proposed empirical test of leverage targeting behavior provides strong evidence in favor of the dynamic trade-off theory.

The article proceeds as follows: Section 2 reviews the relevant literature, summarizes contributions to the literature, sets the research agenda, and discusses the testable hypotheses. Section 3 introduces the data and the empirical method. Sections 4 and 5 present and discuss the results. Section 6 offers conclusions.

## 2. Literature Review and Hypotheses Development

### 2.1. Literature Review

Research since the seminal works of Modigliani and Miller (1958, 1963) and Miller (1977) has tested the predictions of the four pre-eminent theories of capital structure: (i) the static tradeoff model in which firms balance the various costs and benefits of debt (e.g., Kraus and Litzenberger 1973; Jensen and Meckling 1976; Myers 1977; Green 1984; Jensen 1986; Mayers 1998); (ii) the dynamic tradeoff model that suggests a target adjustment process in the presence of adjustment costs (e.g., Hennessy and Whited 2005; Leary and Roberts 2005); (iii) the pecking order model in which firms minimize the adverse selection costs associated with security issuances by adhering to a financing hierarchy where retained earnings are preferred over debt and where debt is preferred over equity (e.g., Donaldson 1961; Myers 1984; Brennan and Kraus 1987; Myers and Majluf 1984; Stein 1992); and (iv) the market timing model where the deviations of security prices from their fundamental values or from their relative costs drive issuance decisions (Baker and Wurgler 2002). All of the theories have some empirical support (e.g., Frank and Goyal 2003; Flannery and Rangan 2006; Lemmon et al. 2008; Frank and Goyal 2009; Huang and Ritter 2009; Öztekin and Flannery 2012; DeAngelo and Roll 2015). Harris and Raviv (1991) and Frank and Goyal (2003), among others, provide excellent surveys. Recently, researchers have documented other factors (e.g., reporting classifications for financing instruments, whether the company is multinational or domestic) affecting firm financing (e.g., Levi and Segal 2015). This paper contributes to the dynamic tradeoff category by proposing a new empirical test for leverage targeting behavior that is based on leverage targets and deviations as opposed to actual leverage.

Empirical tests of target behavior can be summarized under two broad categories. The first category of studies documents the evidence of convergence toward an optimum over time in U.S. and international samples (e.g., Fama and French 2002; Flannery and Rangan 2006; Faulkender et al. 2012; Öztekin and Flannery 2012). Several studies have also examined the rate of adjustment as a function of various firm and country characteristics to shed more light on the nature of adjustment costs and benefits (e.g., Faulkender et al. 2012; Öztekin and Flannery 2012). Overall, these studies indicate that firms actively pursue target debt ratios, though market frictions might lead to temporary deviations from the optimum. Accordingly, the appropriate modeling of leverage ratios should be dynamic to allow for partial adjustment toward the target as opposed to static models that uniformly assume instantaneous and complete adjustment.

This paper relates to this strand in two ways. First, one of the target specifications is dynamic and similarly allows for the partial adjustment of debt ratios. Second, similar to the studies that model the speed of adjustment as a function of various firm and country characteristics, our empirical model follows a two-step procedure. In the initial step, the estimates of targets and target deviations are obtained. In the second step, these estimates are used as key measures to test the underlying hypotheses. However, while the dependent variable is the change in the observed leverage and the independent variables are the interaction terms between the adjustment cost and benefit factors and the target deviations in the aforementioned studies, the dependent variable is either the estimated leverage target or target deviation in the specifications of this paper.

This study contributes to this strand by modeling target deviations as a function of institutional and financial factors and thereby testing, for the first time, a direct implication of an established fact from the existing studies. In environments that are more conducive to faster capital structure adjustments, ceteris paribus, firms would be closer to

their leverage targets. Consequently, all else being equal, it is expected that persistent deviations from the target would be less likely in strong institutional and financial settings and during better times, leading to lower financial frictions preventing firms from undertaking leverage adjustments. Furthermore, the effect of the recent financial crisis on leverage adjustment processes has not been yet studied. Our findings indicate that external financing constraints became more binding during the crisis, leading to a much higher likelihood of target deviations, which was especially the case for financially constrained firms.

The second category of studies relates observed debt ratios to various firm, industry, macroeconomic, and country attributes that could alter the costs and benefits of operating at various leverage ratios and examines whether, in leverage regressions, these determinants have signs that are consistent with the trade-offs. Most of these studies employ static leverage regressions, implicitly assuming that the observed debt ratio is an adequate proxy for the optimal debt ratio (Frank and Goyal 2009; Rajan and Zingales 1995; Booth et al. 2001; Fan et al. 2012). Other studies in this category account for the dynamic nature of leverage data and allow for deviations from the optimal level, but they also continue to focus on testing the determinants of observed debt ratios and solely examine either the total debt ratio or the maturity of debt, but not both (Gungoraydinoglu and Öztekin 2011; Öztekin 2015).

This study contributes to this strand by modeling leverage targets as a function of institutional, financial, and macroeconomic factors (e.g., Özer and Çam 2020). The recent financial crisis (2007–2008) had profound effects around the globe, yet little is known about how it affected the capital structure choices of firms. This study adds to the existing literature by testing the impact of the recent financial crisis on leverage targets for the first time. Furthermore, the current study also analyzes how the recent financial crisis has affected the relationships among leverage targets and a country's institutional strength.

*2.2. Hypotheses Development*

This study employs an alternative way to study the capital structure targeting behavior of firms. The debt ratio targets of firms are the outcomes of an optimization process in which the firm trades off the costs and benefits of debt, as reflected in firm, industry, macroeconomic, and country attributes. If the estimates of the target capital structures are sensible, the cross-sectional tests of the fitted targets and target deviations should reveal these costs and benefits.

An important conjecture underlying the existence of an optimal capital structure is that in the absence of significant adjustment costs, firms strive to minimize deviations between actual and target capital structures to prevent the detrimental valuation effects of suboptimal capital structures. If there were no adjustment costs toward the target, each firm's observed capital structure would also be its optimal capital structure. If, however, firms cannot promptly react to the deviations from their optimal structure, cross-sectional variations in actual capital structures would be able to be observed, even across a sample of firms having the same target capital structures. Similarly, shocks to the financial performance, investment opportunity set, market values of firms as well as other factors could lead to fluctuations in target debt ratios over time without necessarily having the same influence on the observed debt ratios. Therefore, the method of investigating the cross-sectional determinants of target debt ratios is a more realistic method compared to examining the cross-sectional determinants of actual leverage ratios. Our first hypothesis is that the extent to which various cost and benefit factors affect observed leverage ratios as opposed to leverage targets and/or target deviations differ across firms and/or over time.

There are three additional testable predictions of capital structure targeting behavior in the context of the new empirical approach. One prediction posits that the estimated targets should be meaningfully and significantly different across institutional environ-

ments, reflecting the various costs and benefits of financing imposed on firms. For example, stronger shareholder protection tends to reduce the agency costs of equity, making it cheaper for firms to issue equity. If so, stronger shareholder protection should correspond to lower target leverage. Furthermore, because short-term debt mitigates the loss in the firm value attributed to the separation of ownership and control, firms operating in a weak shareholder protection environment may target more short-term debt in their capital structure. More generally, finding that leverage targets vary plausibly with institutional and financial variables would lend support to the hypotheses that firms optimally choose their capital structure. To test our second hypothesis, estimated targets are tested across portfolios that were formed based on the quality of legal and financial institutions.

Another prediction suggests that poor market conditions negatively affect a firm's overall demand for and access to capital by raising external financing costs (Campello et al. 2010; McLean and Zhao 2014). Therefore, the financial crisis should alter the desired structure of the securities offered, shifting them toward shorter maturity. In addition, the adverse effects of market conditions on financing patterns should be moderated in countries with strong legal and financial institutions that subject firms to lower external financing costs (La Porta et al. 1997, 1998). Therefore, according to our third hypothesis, the differences in the targeting behavior of firms in weak and strong institutional environments should become more pronounced during the financial crisis.

Our last prediction conjectures that negative shocks to the supply of finance, in conjunction with the presence of financing frictions, could impede capital structure adjustments and result in a higher likelihood and a greater magnitude of target deviations for extended periods. Assuming that two firms—one operating in a stronger institutional and financial setting and the other operating in a weaker institutional and financial setting—start off with the same leverage deviation and are subject to similar shocks, the firm in the strong institutional environment and/or the well-developed economy would be less likely to deviate from its target capital structure than the one in the weak institutional environment and/or the poorly developed economy due to the lower financial frictions associated with the leverage adjustments in the former. Similarly, financial crises would represent a significant negative shock to the supply and the cost of external finance for firms (Bernanke et al. 1996), leading to a greater likelihood of target deviations. Furthermore, theory and empirical evidence suggest that the effect of such an external shock should be more severe for financially constrained firms that face relatively higher external capital costs (Bernanke et al. 1996; Kiyotaki and Moore 1997).

In summary, our discussion leads to the following hypotheses:

**Hypothesis 1 (H1).** *The extent to which various cost and benefit factors affect observed leverage ratios as opposed to leverage targets and/or target deviations differ across firms over time.*

**Hypothesis 2 (H2).** *Estimated targets are meaningfully and significantly different across institutional environments, reflecting the various costs and benefits of financing imposed on firms.*

**Hypothesis 3 (H3).** *The differences in the targeting behavior of firms in weak and strong institutional environments are more pronounced during a financial crisis.*

**Hypothesis 4 (H4).** *Financial crises are associated with a greater likelihood of target deviations, particularly for the financially constrained firms that face higher external financing costs.*

### 3. Sample and Methods

*3.1. Sample*

To examine the targeting behavior of firms, this study relies on the Compustat Global Vantage database and excludes financial firms (Standard Industrial Classification [SIC] 6000–6999) and utilities (SIC 4900–4999). The sample covers the period of 1989–2017. Firm-level data were collected for firms for which information about country-level institutional

features could be located. To alleviate short-panel bias, firms that report information for fewer than five consecutive years were deleted. In addition, to attain a reasonable cross-sectional variation within each country, countries that have fewer than ten firms reporting the required accounting data were eliminated. The final sample is an unbalanced panel with 17,193 firms from 46 countries, for a total of 183,124 firm years. All variables were winsorized at the 1% and 99% levels to mitigate the impact of outliers. The sample consists of a mix of developed and developing countries from Europe and the rest of the world, including U.S. Panel A of Table 1 provides information on the definition, source, construction, and summary statistics of the firm, industry, macroeconomic, and country-level variables that are used to explain the variation in target capital structures. Panel B documents the correlations among institutional variables. An average (median) firm in the sample has a total debt ratio of 23.49% (20.84%) and a long-term debt ratio of 14.24% (9.53%).

**Table 1.** Data description, summary statistics, and correlations.

**Panel A. Definitions, Source, and Summary Statistics of Variables**

| *Variable: Definition (Factor Loading in the Principal Component Analysis) (Source)* | Mean | Median | Std. Dev. |
|---|---|---|---|
| Total book debt ratio: (Long-term debt+Short-term debt)/Total assets (Global Vantage). | 0.2349 | 0.2084 | 0.1983 |
| Long-term book debt ratio: Long-term debt/Total assets (Global Vantage). | 0.1424 | 0.0953 | 0.1571 |
| Profit: (Operating Income+Interest and related expense+Current Income Taxes)/Total assets (Global Vantage). | 0.0352 | 0.0617 | 0.1803 |
| Market-to-book: (Long-term debt+Short-term debt+Preferred capital+Market value of equity)/Total assets (Global Vantage). | 1.1904 | 0.8380 | 1.2686 |
| Depreciation: Total Depreciation and Amortization/Total assets (Global Vantage). | 0.0436 | 0.0369 | 0.0322 |
| Size: Log[Lagged sales]/Sales (Global Vantage). | 0.0035 | 0.0091 | 0.4345 |
| Tangibility: Fixed assets/Total assets (Global Vantage). | 0.3102 | 0.2738 | 0.2162 |
| R&D dummy: A dummy variable equal to 1 if R&D expenditures are not reported and 0 if otherwise (Global Vantage). | 0.3941 | 0.0000 | 0.4887 |
| R&D expenses: R&D expense/Total assets (Global Vantage). | 0.0228 | 0.0000 | 0.0596 |
| Industry median total book debt ratio: Median total book debt ratio for the firm's industry (Global Vantage). | 0.2027 | 0.2044 | 0.0966 |
| Industry median long-term book debt ratio: Median total long-term book debt ratio for the firm's industry (Global Vantage). | 0.0843 | 0.0924 | 0.0344 |
| Dividend payer: A dummy variable equal to 1 if the firm has paid dividends and 0 if otherwise (Global Vantage). | 0.7605 | 1.0000 | 0.4268 |
| Capital expenditures: Capital expenditures/Total assets (Global Vantage). | 0.0538 | 0.0359 | 0.0591 |
| Stock market capitalization: The value of listed shares divided by GDP (Beck et al. 2000). | 0.9767 | 0.9692 | 0.4359 |
| Bond market capitalization: Outstanding domestic debt securities divided by GDP (Beck et al. 2000). | 0.6107 | 0.4871 | 0.4526 |
| Private credit: Private credit by deposit money banks and other financial institutions to GDP (Beck et al. 2000). | 0.8452 | 0.8308 | 0.4042 |
| Crisis: A dummy variable equal to 1 for the years 2007 and 2008; 0 if otherwise (Global Vantage). | 0.1334 | 0.0000 | 0.3400 |
| Legal origin: Categorical variables (for common law and civil law) equal to unity if the firm operates under the named legal origin, and zero if otherwise (La Porta et al. 1998). | 0.3868 | 0.0000 | 0.4870 |
| Financial development: Dummy variable equal to unity if the financial system is developed and 0 if otherwise (Levine 2002). | 0.9234 | 1.0000 | 0.2660 |
| Financial structure: Dummy variable equal to 1 if the financial structure is market-based and 0 if otherwise (Levine 2002). | 0.9158 | 1.0000 | 0.2777 |
| Shareholder protection: Principal component of shareholder rights (0.88) (La Porta et al. 1998) and their enforcement (0.88) (Djankov et al. 2008a). | 0.0000 | 0.5423 | 1.0000 |
| Internal governance: Principal component of executive constraints (0.73) (Djankov et al. 2002) and corporate governance (0.73) (World Bank). | 0.0000 | 0.3953 | 1.0000 |
| External governance: Principal component of corruption (0.86), risk of expropriation (0.92), repudiation (0.89) by government (La Porta et al. 1998); law and order (0.95); and enforceability of contracts (0.88) (Djankov et al. 2003). | 0.0000 | 0.5347 | 1.0000 |
| Tax benefits: Statutory tax rate (Djankov et al. 2008b). | 0.3759 | 0.4210 | 0.7788 |
| Bankruptcy procedures: Principal component of time (−0.85), cost (−0.85), and efficiency (0.92) to resolve insolvency (Djankov et al. 2008b). | 0.0000 | −0.0405 | 1.0000 |
| Creditor protection: Principal component of creditor rights (0.30) (La Porta et al. 1998) and their enforcement (0.30) (Djankov et al. 2003). | 0.0000 | −0.1249 | 1.0000 |

Ownership and control: Principal component of widely held firm (−0.56), family firm (0.53), and state firm (0.84) variables. A firm is treated as having a controlling shareholder if the sum of a shareholder's direct and indirect voting rights exceeds 20%. Firms are defined as widely held if they have no controlling shareholder; firms are defined as family firms if a person is the controlling shareholder; firms are defined as and state firms if a domestic or foreign state is the controlling shareholder (La Porta et al. 1999). The widely held firm, family firm, and state firm variables denote the fraction of each type of firm in a given country.　　0.0000　　−0.3054　　1.0000

Equity information asymmetry: Principal component of accounting standards (0.73) (La Porta et al. 1998), disclosure requirements (0.86), liability standards (0.81), public enforcement (0.79), and the prevalence of insider trading (0.64) (La Porta et al. 2006).　　0.0000　　0.4217　　1.0000

Debt information asymmetry: Principal component of accounting standards (0.76) and the public credit registry (0.76) (Djankov et al. 2007).　　0.0000　　0.4052　　1.0000

**Panel B. Correlation Matrix for the Institutional Indices**

|  | Legal Origin | Financial Development | Financial Structure | Shareholder Protection | Internal Governance | External Governance | Tax Benefits | Bankruptcy Procedures | Creditor Protection | Ownership and Control | Equity Information Asymmetry | Debt Information Asymmetry |
|---|---|---|---|---|---|---|---|---|---|---|---|---|
| Legal Origin | 1.0000 | | | | | | | | | | | |
| Financial Development | −0.2876 | 1.0000 | | | | | | | | | | |
| Financial Structure | −0.3950 | 0.1388 | 1.0000 | | | | | | | | | |
| Shareholder Protection | −0.5712 | 0.3862 | 0.1070 | 1.0000 | | | | | | | | |
| Internal Governance | −0.1314 | 0.3706 | 0.1627 | 0.1438 | 1.0000 | | | | | | | |
| External Governance | −0.1210 | 0.2811 | 0.0974 | 0.0393 | 0.7321 | 1.0000 | | | | | | |
| Tax Benefits | 0.0963 | 0.0936 | 0.0949 | 0.0402 | 0.0714 | 0.0625 | 1.0000 | | | | | |
| Bankruptcy Procedures | −0.1129 | 0.3459 | 0.0692 | 0.0673 | 0.6455 | 0.6682 | 0.3405 | 1.0000 | | | | |
| Creditor Protection | −0.3974 | 0.2892 | 0.3547 | 0.4135 | 0.3188 | 0.1022 | 0.0933 | 0.2156 | 1.0000 | | | |
| Ownership and Control | 0.3653 | −0.1293 | −0.2984 | −0.4485 | −0.1196 | −0.0091 | −0.0262 | −0.0612 | −0.3258 | 1.0000 | | |
| Equity Information Asymmetry | −0.3444 | 0.3894 | 0.3797 | 0.2607 | 0.3783 | 0.3393 | 0.0625 | 0.1116 | 0.2054 | −0.4193 | 1.0000 | |
| Debt Information Asymmetry | −0.3648 | 0.3258 | 0.2284 | 0.3267 | 0.5345 | 0.2547 | 0.0012 | 0.3333 | 0.2673 | −0.1576 | 0.6446 | 1.0000 |

*3.2. Methods*

Three different empirical approaches were used to estimate the target capital structures: Blundell and Bond's (1998) approach, Fama and French's (2002) approach, and the median industry leverage in each country. The selection of the firm- and industry-specific variables that describe target leverage was based on the previous literature (Flannery and Rangan 2006; Frank and Goyal 2009). Accordingly, each firm's optimal leverage ($LEV_{ij,t}^{*}$) varies over time and according to its characteristics, $X$:

$$LEV_{ij,t}^{*} = \beta X_{ij,t-1} \tag{1}$$

where

$LEV_{ij,t}^{*}$ is the $i$th firm's target leverage for the end of the year $t$ in country $j$;

$\beta$ is a coefficient vector;

$X_{ij,t-1}$ is a vector of the firm and industry characteristics for the $i$th firm in country $j$ for the end of the year $t-1$ and includes earnings before interest and taxes as a proportion

of the total assets; book liabilities plus the market value of equity to the total assets; depreciation expense as a proportion of the total assets; log sales scaled with the total sales (a measure of firm size); fixed assets as a proportion of the total assets; research and development expenses as a proportion of the total assets, where missing values are captured with a binary variable, and the median leverage ratio for the firm's industry, based on the Fama and French (1997) industry categories, where leverage is either measured by the total (book) debt to the total assets or by the total long-term debt to the total assets.

The first approach to estimate the targets is based on the work of Blundell and Bond (1998) and employs a system-generalized method of moments (GMM) (Flannery and Hankins 2013). Confronted with a set of costs and benefits when rebalancing capital structure, managers will choose how quickly to close any gap between the actual and target capital structure. The adjustment speed ($\lambda$) permits a typical firm to move only part of the way toward its target leverage for any given year. If managers have target debt ratios and make proactive efforts to reach them, $\lambda$ should be strictly greater than zero. In the presence of market frictions, the adjustment is not instantaneous, so $\lambda$ is less than one. A partial adjustment specification is used to estimate leverage targets:

$$LEV_{ij,t} = (\lambda\beta)X_{ij,t-1} + (1-\lambda)LEV_{ij,t-1} + \delta_{ij,t} \tag{2}$$

A two-step system GMM is used to estimate Equation (2) and includes firm-, year-, and country-fixed effects to control for unobservable factors that could affect leverage. The potential endogeneity of the right-hand-side variables is controlled by the lags of the same variables as instruments. Using the actual $LEV_{ij,t}$, $LEV_{ij,t-1}$, and the estimated speed of adjustment $\lambda$, the targets are then extracted as the predicted value of the following equation:

$$\beta X_{ij,t-1} = \frac{1}{\hat{\lambda}}\left(LEV_{ij,t} - \left(1-\hat{\lambda}\right)LEV_{ij,t-1}\right) + \frac{\hat{\delta}}{\hat{\lambda}_{ij,t}}. \tag{3}$$

The second approach to estimate the targets is based on the work of Fama and French (2002) and employs Fama and MacBeth's (1973) cross-sectional leverage regressions of Equation (1) estimated year-by-year with the inclusion of the country-fixed effects. The predicted values from these annual regressions were then used as the target estimates. To draw inferences about the average slopes, Fama and MacBeth's time-series standard errors adjusted for cross-correlation and autocorrelation were computed. Consequently, similar to Blundell and Bond (1998), this estimation technique powerfully addresses econometric challenges, including the cross-correlation and inference problems that are potentially caused by autocorrelation in the regression residuals.

Finally, many studies raise the possibility that managers use industry median leverage as a benchmark, as they choose their own firm's target leverage (e.g., Flannery and Rangan 2006; Frank and Goyal 2009). Accordingly, in addition to Blundell and Bond's (Blundell and Bond 1998) and Fama and French's (Fama and French 2002) methodologies, a simple industry median leverage ratio was computed in each country using both total debt and long-term debt ratios as a third alternative. Although variants of the target estimations are presented, none of the methodologies adopted in this study are assumed to be superior to the others, and the use of three different empirical approaches is mainly for robustness reasons.

The impact of the institutional and financial environment in explaining international differences in leverage targets are captured with the inclusion of country and time fixed effects in the estimation (fitting) of the leverage targets. Similar results are obtained if country-level observables are directly included in the target estimation. We report the results with the country-fixed effects since this methodology is less susceptible to model misspecification and omits variable problems.

The target estimations employ the total debt ratio ($LEV^T$) and the total long-term debt ratio ($LEV^{LT}$):

$$LEV^T = \frac{Long\ Term\ Debt + Short\ Term\ Debt}{Total\ Assets}, \tag{4}$$

$$LEV^{LT} = \frac{Long\ Term\ Debt}{Total\ Assets}. \tag{5}$$

Table 2 presents the estimations of Equation (3) from Blundell and Bond's (1998) approach and the average annual slope coefficient estimates from Fama and French's (2002) approach. The coefficient estimates are consistent across the two methodologies and are in line with findings from previous research. Firms with higher profits, market-to-book ratios, depreciation, and research-and-development expenses desire lower leverage and long-term debt; larger firms, firms that have more tangible assets, and firms that compete in industries in which the median firm has higher debt ratios target higher leverage and long-term debt.

**Table 2.** Target capital structure estimations.

| | Total Debt Ratio | | Long-Term Debt Ratio | |
|---|---|---|---|---|
| | **Blundell and Bond (1998)** | **Fama and French (2002)** | **Blundell and Bond (1998)** | **Fama and French (2002)** |
| Leverage | 0.8200 *** | | 0.6781 *** | |
| | (0.000) | | (0.000) | |
| Profit | −0.0138 *** | −0.2031 *** | −0.0146 *** | −0.0737 *** |
| | (0.004) | (0.000) | (0.000) | (0.000) |
| Market-to-book | −0.0026*** | −0.0002 | −0.0020 *** | 0.0012 |
| | (0.000) | (0.919) | (0.000) | (0.410) |
| Depreciation | −0.1643 *** | −0.1021 ** | −0.1006 *** | −0.0949 ** |
| | (0.000) | (0.021) | (0.000) | (0.045) |
| Size | −0.0010 | 0.0188 *** | 0.0026 | 0.0099*** |
| | (0.617) | (0.000) | (0.106) | (0.000) |
| Tangibility | 0.0603 *** | 0.1379 *** | 0.0531 *** | 0.1992 *** |
| | (0.000) | (0.000) | (0.000) | (0.000) |
| R&D dummy | −0.0029 | −0.0015 | −0.0010 | −0.0156*** |
| | (0.168) | (0.618) | (0.554) | (0.000) |
| R&D expenses | −0.0197 | −0.3076 *** | −0.0328 * | −0.2442 *** |
| | (0.372) | (0.000) | (0.071) | (0.000) |
| Median industry leverage | 0.0713 *** | 0.5478 *** | −0.1410 | 1.0315 *** |
| | (0.000) | (0.000) | (0.719) | (0.000) |
| Constant | 0.0243 *** | 0.1080 *** | 0.0405 ** | 0.0150 *** |
| | (0.000) | (0.000) | (0.024) | (0.000) |
| Observations | 183,071 | 183,071 | 183,124 | 183,124 |
| R-Squared | 0.780 | 0.163 | 0.490 | 0.129 |
| F | 349.42 | 1038.69 | 358.976 | 938.61 |
| Prob > F | 0.000 | 0.000 | 0.000 | 0.000 |

Notes: The table presents target capital structure estimations for book leverage using Blundell and Bond's (1998) and Fama and French's (2002) approaches during the sample period (1989–2017) for the total debt ratio and long-term debt ratio, respectively. The definitions of the variables in the regressions appear in Table 1. All of the variables used in the estimations are lagged by one period to avoid reverse causality. The first and the third columns report the estimation results from Blundell and Bond's (Blundell and Bond 1998) two-step system GMM with the inclusion of the unreported firm-, year-, and country-fixed effects. The second and the last columns present the average annual slope coefficient estimates from Fama and French's (2002) approach in which Fama and MacBeth's (1973) cross-sectional regressions are estimated for each year and include country-fixed effects. The reported standard errors in Fama and French's (2002) are the standard deviation of the n slope estimates divided by $\sqrt{n}$. The targets are the predicted values from these regressions, are time-varying, and incorporate both firm-, industry-, and country-level information. The top row reports the coefficient estimates. The *p*-values of the two tailed tests for the significance of the coefficients appear beneath the coefficient estimates in parentheses. *, **, and *** indicate significance at the 10%, 5%, and 1% levels. R-Squared is the proportion of variance in the dependent variable, which can be explained by the independent variables. F is the F-statistic testing the null hypothesis that all of the model coefficients are zero. Prob > F is the p-value associated with the F-statistic.

In this paper, we argue that the extent to which various cost and benefit factors affect thre observed leverage ratios as opposed to the leverage targets may vary across firms over time. To examine the relative importance of institutional and financial development factors in explaining the observed leverage of firms as opposed to their estimated leverage target, we estimate the following reduced model:

$$DEP_{ij,t} = \mu + \pi Z_{ij,t-1} + \xi_{ij,t-1}, \tag{6}$$

where *i* indexes the firms, *j* indexes the countries, *t* indexes the years, the dependent variable is either the observed leverage or the target leverage: $DEP_{ij,t} = \{LEV_{ij,t} \ or \ LEV^*_{ij,t}\}$, $\mu$ is the average response across all firms, $Z$ is a set of institutional and financial development variables, and $\xi$ is a random error term. The relative importance of the institutional and financial development factors for the observed leverage as opposed to those for the target leverage can be inferred by the coefficient estimate of $\pi$ obtained from separate estimations of Equation (6) for the observed and the target leverage.

## 4. Results

### 4.1. Leverage Targets

Table 3 documents leverage targets for a comprehensive international sample. The results are documented for the three methods used to estimate the targets: Blundell and Bond's (1998) two-step system GMM (Target 1), Fama and French's (2002) cross-sectional leverage regressions estimated annually (Target 2), and the median industry leverage in each country (Target 3). Mean statistics are provided for the actual leverage, the estimates of the target leverage, and the *t*-tests of the difference between the estimated targets and the observed capital structures separately for each of the 46 countries and the overall sample during the 1989–2017 period as well as the three sub periods: before, during, and after the global financial crisis.

**Table 3.** Actual and target capital structures.

| | Countries | Panel Structure | | Total Debt Ratio (%) | | | | Long-Term Debt Ratio (%) | | | |
|---|---|---|---|---|---|---|---|---|---|---|---|
| | | Firms | Average | Actual | Target 1 | Target 2 | Target 3 | Actual | Target 1 | Target 2 | Target 3 |
| 1 | Argentina | 35 | 9 | 21.14 | 22.16 | 24.30 *** | 17.77 *** | 11.27 | 11.24 | 14.78 *** | 7.36 *** |
| 2 | Australia | 774 | 9 | 18.67 | 21.31 *** | 20.68 *** | 13.77 *** | 12.31 | 12.91 *** | 7.20 *** | 8.28 *** |
| 3 | Austria | 62 | 11 | 24.62 | 28.83 *** | 25.35 | 23.37 ** | 13.16 | 14.38 *** | 17.24 *** | 11.78 *** |
| 4 | Belgium | 85 | 11 | 24.28 | 21.89 *** | 25.29 *** | 23.40 * | 14.65 | 14.33 | 17.94 *** | 13.24 *** |
| 5 | Brazil | 66 | 7 | 27.24 | 28.89** | 25.97 | 23.11 *** | 13.58 | 13.89 | 18.11 *** | 10.61 *** |
| 6 | Canada | 189 | 10 | 22.32 | 23.72 *** | 24.97 *** | 20.27 *** | 17.24 | 17.75 ** | 16.48 ** | 14.81 *** |
| 7 | Chile | 38 | 8 | 23.60 | 23.32 | 28.46 *** | 23.59 | 15.03 | 15.99 ** | 23.05 *** | 14.22 |
| 8 | Denmark | 103 | 11 | 27.07 | 27.74 | 26.95 | 25.66 *** | 15.98 | 16.13 | 19.19 *** | 14.57 *** |
| 9 | Finland | 114 | 11 | 25.63 | 26.05 | 24.90 * | 23.48 *** | 17.31 | 16.98 | 20.87 *** | 14.84 *** |
| 10 | France | 502 | 10 | 22.81 | 22.57 | 22.85 | 21.38 *** | 13.50 | 13.44 | 14.18 *** | 10.78 *** |
| 11 | Germany | 536 | 10 | 20.45 | 21.66 *** | 21.16 *** | 17.30 *** | 11.39 | 12.06 *** | 11.04 ** | 8.08 *** |
| 12 | Greece | 100 | 8 | 32.82 | 41.00 *** | 31.52 ** | 32.99 | 16.60 | 18.84 *** | 20.50 *** | 14.36 *** |
| 13 | India | 550 | 7 | 30.11 | 29.99 | 29.85 | 30.13 | 19.84 | 19.28 *** | 23.33 *** | 17.45 *** |
| 14 | Indonesia | 179 | 10 | 38.19 | 36.53 *** | 32.71 *** | 34.95 *** | 17.48 | 17.50 | 18.46 ** | 12.80 *** |
| 15 | Ireland | 33 | 11 | 25.54 | 25.74 | 27.71 *** | 24.53 | 18.19 | 18.66 | 19.93 ** | 16.46 *** |
| 16 | Israel | 23 | 9 | 37.73 | 39.52 * | 31.32 *** | 37.52 | 25.92 | 27.92 *** | 13.79 *** | 25.67 |
| 17 | Italy | 168 | 10 | 25.85 | 28.95 *** | 26.18 | 25.62 | 13.65 | 14.77 *** | 15.60 *** | 11.55 *** |
| 18 | Japan | 2848 | 11 | 23.83 | 22.35 *** | 26.54 *** | 21.07 *** | 10.29 | 10.02 *** | 14.91 *** | 7.76 *** |
| 19 | Latvia | 10 | 7 | 17.94 | 20.97* | 24.44 *** | 16.08 | 11.02 | 11.91 | 10.89 | 9.90 |
| 20 | Lithuania | 19 | 6 | 26.09 | 25.23 | 29.83 *** | 26.60 | 13.89 | 11.84 ** | 23.50 *** | 10.93 *** |
| 21 | Luxemburg | 11 | 8 | 21.93 | 19.69 ** | 27.05 *** | 20.61 | 15.75 | 15.63 | 23.39 *** | 15.91 |
| 22 | Malaysia | 739 | 10 | 24.41 | 26.99 *** | 25.56 *** | 21.36 *** | 8.88 | 9.08 * | 11.59 *** | 4.67 *** |
| 23 | Mexico | 57 | 10 | 26.71 | 27.56 * | 28.85 *** | 26.21 | 18.14 | 18.78 * | 24.53 *** | 18.25 |
| 24 | Netherlands | 128 | 12 | 22.18 | 23.86 *** | 22.99 ** | 19.95 *** | 13.00 | 13.73 *** | 16.72 *** | 10.80 *** |
| 25 | New Zealand | 67 | 10 | 22.92 | 23.55 | 26.48 *** | 21.86 * | 15.79 | 15.34 | 21.20 *** | 14.35 *** |
| 26 | Nigeria | 11 | 7 | 22.00 | 25.04 * | 20.66 | 18.48 ** | 4.74 | 4.85 | 8.58 *** | 2.66 ** |
| 27 | Norway | 113 | 10 | 27.37 | 27.73 | 28.31 ** | 27.45 | 21.35 | 21.19 | 25.61 *** | 20.41 ** |
| 28 | Pakistan | 83 | 9 | 30.98 | 29.43 *** | 30.69 | 29.77 ** | 14.04 | 13.59 | 18.04 *** | 12.00 *** |
| 29 | Peru | 25 | 9 | 22.20 | 19.07 *** | 25.95 *** | 19.76 *** | 11.46 | 10.36 ** | 17.76 *** | 8.80 *** |

| 30 | Philippines | 66 | 10 | 26.14 | 26.02 | 25.79 | 23.29 *** | 14.08 | 14.78 * | 11.02 *** | 10.51 *** |
| 31 | Poland | 112 | 8 | 18.12 | 21.45 *** | 21.28 *** | 15.58 *** | 7.58 | 8.90 *** | 9.82 *** | 5.16 *** |
| 32 | Portugal | 43 | 10 | 36.68 | 41.38 *** | 32.74 *** | 36.05 | 21.28 | 23.22 *** | 26.68 *** | 20.13 * |
| 33 | Russian Federation | 15 | 7 | 27.27 | 34.05 *** | 27.81 | 27.05 | 17.70 | 21.18 *** | 19.96 * | 17.34 |
| 34 | Singapore | 386 | 10 | 20.49 | 20.30 | 23.80 *** | 17.70 *** | 8.65 | 8.42 * | 10.59 *** | 5.03 *** |
| 35 | Slovenia | 15 | 9 | 26.47 | 34.69 *** | 28.07 | 26.54 | 12.32 | 15.17 *** | 20.00 *** | 10.53 ** |
| 36 | South Africa | 175 | 10 | 16.84 | 19.58 *** | 19.08 *** | 13.98 *** | 9.49 | 10.45 *** | 10.41 *** | 6.22 *** |
| 37 | South Korea | 366 | 10 | 31.97 | 29.47 *** | 32.32 | 31.75 | 12.29 | 11.40 *** | 17.96 *** | 9.86 *** |
| 38 | Spain | 99 | 11 | 24.89 | 26.89 *** | 27.05 *** | 23.90 ** | 14.41 | 15.28 *** | 18.12 *** | 12.16 *** |
| 39 | Sri Lanka | 25 | 8 | 25.36 | 22.30 *** | 28.45 *** | 24.83 | 10.32 | 9.52 ** | 16.78 *** | 9.07 *** |
| 40 | Sweden | 245 | 10 | 19.31 | 20.02 *** | 20.85 *** | 16.85 *** | 13.27 | 13.09 | 11.82 *** | 10.29 *** |
| 41 | Switzerland | 167 | 12 | 22.74 | 19.90 *** | 24.86 *** | 21.41 *** | 15.63 | 14.58 *** | 20.01 *** | 13.24 *** |
| 42 | Taiwan | 743 | 7 | 22.52 | 21.89 *** | 24.60 *** | 21.67 *** | 8.53 | 8.26 *** | 9.57 *** | 5.64 *** |
| 43 | Thailand | 335 | 10 | 28.61 | 26.81 *** | 29.92 *** | 26.10 *** | 11.81 | 11.85 | 13.34 *** | 7.58 *** |
| 44 | Turkey | 58 | 8 | 20.23 | 20.97 | 21.70 ** | 18.34 *** | 9.38 | 10.17 ** | 10.47 ** | 7.00 *** |
| 45 | United Kingdom | 1299 | 11 | 18.96 | 20.72 *** | 21.42 *** | 16.30 *** | 11.81 | 12.58 *** | 10.87 *** | 8.10 *** |
| 46 | United States | 5376 | 12 | 23.58 | 24.91 *** | 22.68 *** | 19.97 *** | 18.25 | 18.50 *** | 15.88 *** | 14.00 *** |
| 47 | 1989–2017 | 2905 | 11 | 23.50 | 24.13 *** | 24.25 *** | 20.72 *** | 14.24 | 14.41 *** | 14.69 *** | 10.83 *** |
| 48 | 1989–2006 | 2668 | 11 | 23.99 | 24.29 *** | 24.22 *** | 20.61 *** | 14.85 | 14.98 *** | 15.01 *** | 11.20 *** |
| 49 | 2007–2008 | 1989 | 10 | 22.29 | 26.78 *** | 23.93 *** | 20.89 *** | 12.55 | 13.14 *** | 13.66 *** | 9.75 *** |
| 50 | 2009–2017 | 2186 | 10 | 22.23 | 21.12 *** | 24.67 *** | 21.07 *** | 12.87 | 12.86 | 14.10 *** | 10.04 *** |

Notes: The table summarizes the actual and target capital structures of firms. Rows 1–46 provide information on the averages for each country in the sample. Rows 47–50 provide the summary statistics of these means for the overall sample and three subperiods: 1989–2006, 2007–2008, and 2009–2017. The panel structure column reports the number of firms and the mean number of observations per firm. The remaining columns report the observed leverage and calculated leverage targets (%) in each country, using book leverage, for total debt ratio and long-term debt ratio, respectively. Target 1 is estimated using Blundell and Bond's (1998) two-step system GMM. Target 2 is obtained from Fama and French's (2002) cross-sectional leverage regressions estimated annually (Fama and MacBeth 1973). Target 3 denotes the median industry leverage in each country. The significance levels from the *t*-tests testing whether the means of calculated targets are equal to the actual leverage appear next to each target estimate. *, **, and *** indicate significance at the 10%, 5%, and 1% levels.

Leverage targets vary considerably across countries, ranging from 13.77% (Australia) to 41.38% (Portugal) using total debt and from 2.66% (Nigeria) to 27.92% (Israel) using long-term debt, respectively. Conditional on the target estimation models, this finding is consistent with the view that the net costs or benefits of financing vary across countries. The deviations of the actual leverage from the estimated leverage targets, which are generally statistically significant at the 1% level, represent absolute differences up to 6.5 (17.94% vs. 24.44% in Latvia) and 3.84 (4.74% vs. 8.58% in Nigeria) percentage points, corresponding to differences of 36.23% (24.44%/17.94%-1) and 81.01% (8.58%/4.74%-1) relative to actual leverage, for total and long-term debt, respectively.

For the full sample period and across all 46 countries, the mean leverage targets range from 20.72–24.25% compared to the observed mean debt ratio of 23.50%, while the means of the long-term leverage targets range from 10.83–14.69% compared to the actual mean long-term debt ratio of 14.24%. The differences are significant at the 1% level using both the total and long-term debt ratios. The economic magnitudes of the target capital structure deviations relative to the actual capital structures are nontrivial, with an average absolute difference ranging from 0.63–2.78 percentage points (23.50% vs. 24.13% and 20.72%) translating into a relative difference ranging from 2.68–11.83% for the total debt ratio and a difference ranging from 0.17–3.41 percentage points in absolute terms (14.24% vs. 14.41% and 10.83%) and 1.19–23.95% in relative terms with the long-term debt ratio, depending on the target estimation methodology. Deviations became wider during the crisis compared to the period before the crisis, increasing from an average of 1.30 and 1.31 percentage points across the three target estimation methods to 2.51 and 1.50 percentage points, translating into relative changes of 93.08% and 14.50% for total and long-term debt ratios, respectively.

## 4.2. International Variations in Target Capital Structures

The tests that have been conducted so far generally indicate that capital structure targets are different from the actual leverage, both overall in this large international sample and separately in the individual countries. Table 2 documents the speed of adjustment to target capital structure as 18% and 32%, representing total and long-term debt, respectively. This indicates that firms converge to their leverage targets, though not instantaneously.

Since observed capital structures are related to the underlying target capital structures, by extension, the observed and target capital structures should be systematically related to the same underlying factors. Recent literature (e.g., La Porta et al. 1997, 1998) implies that institutional and financial structures vary across countries and that this systematically affects capital structure management of firms. To provide further insight into the theory of optimal capital structure, the choices of target leverage are examined across a large panel of countries. The benefit of examining not only the total debt ratio but also the long-term debt ratio is that the theory has different implications with regard to different types of debt instruments. The benefit of studying leverage choices with an international scope is that the costs and benefits of optimal financing should plausibly vary with the institutional, legal, and financial environment.

An empirical challenge for testing institutional effects on target capital structures is to effectively summarize the relevant information in the indices. We use binary variables to capture the legal (civil vs. common) and financial (bank- vs. market-based and developed vs. developing) traditions. As drawing conclusions solely based on univariate analyses could be misleading, we also employed the first principal component of the related sub-indices to capture specific institutional dimensions (governance mechanisms, investor protection, bankruptcy procedures, tax status, and information asymmetry). This approach effectively allows country characteristics to interact and act as substitutes or complements and converts the binary analysis to a multivariate analysis by transforming the possibly correlated individual country indices into a composite index. Table 1 provides details on the procedure for combining the institutional indices.

Table 4 reports the variance decompositions of leverage from the estimation of Equation (6). Each row reports the fractions of the model sum of squares attributable to each institutional and financial development variable. Specifically, we report the Type III partial sum of squares that is normalized by the sum across the effects. Figure 1 provides a visual representation of these variance decomposition analyses. For brevity, we discuss the results on actual leverage and Target 1, but similar heterogeneities in the relative relevance of the institutional and financial development factors are also observed using alternative target measures.

There is a great deal of heterogeneity in the relative importance of the institutional and financial development indicators between the observed leverage and the target leverage. In Panel A, ownership and control, financial structure, and legal origin are some of the least important determinants for observed leverage, yet they list among the top five for target leverage. For example, legal origin only explains 3% of the variation in total debt but is able to capture 36% of the variation in total debt target. On the other hand, bankruptcy procedures, internal governance, and external governance are listed among the top four factors for the observed leverage, yet they list in the bottom four factors for the target leverage.

There are more commonalities in the determinants of long-term debt and long-term debt targets. In Panel B, legal origin, financial development, external governance, and creditor protection are important determinants for both the actual and target long-term debt ratios. However, while taxes are important for observed long-term debt determination, ranking fifth and explaining 9% of its total variation, taxes rank tenth and only capture 1% of the variation in target long-term debt. Internal governance, on the other hand, is not an important determinant for observed long-term debt (ranks eleventh), explaining

only 1% of its total variation, but ranks fourth for target long-term debt, capturing 7% of its total variation.

In summary, the results indicate important differences in the relative importance of institutional and financial development determinants in explaining the total variation in actual and target leverage, both in terms of their magnitudes as well as their rankings, consistent with **Hypothesis 1** (**H1**). These findings lend support to the idea that testing the cross-sectional determinants of target capital structures is distinct from examining the cross-sectional determinants of observed leverage ratios.

**Table 4.** Variance decomposition analyses of leverage: financial development of institutions .

| Panel A: Total Debt | | | | |
|---|---|---|---|---|
| **Institutions and Financial Development** | **Actual** | **Target 1** | **Target 2** | **Target 3** |
| Legal Origin | 0.03 | 0.36 | 0.24 | 0.02 |
| Financial Development | 0.06 | 0.13 | 0.15 | 0.01 |
| Financial Structure | 0.02 | 0.11 | 0.00 | 0.03 |
| Shareholder Protection | 0.02 | 0.03 | 0.00 | 0.09 |
| Internal Governance | 0.07 | 0.00 | 0.17 | 0.14 |
| External Governance | 0.09 | 0.00 | 0.01 | 0.10 |
| Tax Benefits | 0.52 | 0.29 | 0.13 | 0.40 |
| Bankruptcy procedures | 0.06 | 0.01 | 0.10 | 0.08 |
| Creditor Protection | 0.04 | 0.02 | 0.12 | 0.01 |
| Ownership and Control | 0.00 | 0.03 | 0.04 | 0.00 |
| Equity Information Asymmetry | 0.05 | 0.01 | 0.04 | 0.08 |
| Debt Information Asymmetry | 0.04 | 0.02 | 0.00 | 0.05 |
| Panel B: Long-Term Debt | | | | |
| **Institutions and Financial Development** | **Actual** | **Target 1** | **Target 2** | **Target 3** |
| Legal Origin | 0.12 | 0.33 | 0.25 | 0.02 |
| Financial Development | 0.13 | 0.18 | 0.18 | 0.13 |
| Financial Structure | 0.05 | 0.02 | 0.00 | 0.09 |
| Shareholder Protection | 0.02 | 0.00 | 0.00 | 0.06 |
| Internal Governance | 0.01 | 0.07 | 0.03 | 0.01 |
| External Governance | 0.13 | 0.03 | 0.01 | 0.10 |
| Tax Benefits | 0.09 | 0.01 | 0.03 | 0.09 |
| Bankruptcy procedures | 0.09 | 0.02 | 0.02 | 0.04 |
| Creditor Protection | 0.30 | 0.30 | 0.42 | 0.37 |
| Ownership and Control | 0.00 | 0.01 | 0.01 | 0.00 |
| Equity Information Asymmetry | 0.04 | 0.02 | 0.05 | 0.06 |
| Debt Information Asymmetry | 0.02 | 0.00 | 0.00 | 0.02 |

Notes: The table reports the variance decompositions of the observed and target leverage from Equation (6). Each row reports the fractions of the model sum of squares attributable to each institutional and financial development variable. Target 1 was estimated using Blundell and Bond's (1998) two-step system GMM. Target 2 was obtained from Fama and French's (2002) cross-sectional leverage regressions estimated annually (Fama and MacBeth 1973). Target 3 denotes the median industry leverage in each country. The definitions and the sources of the variables appear in Table 1.

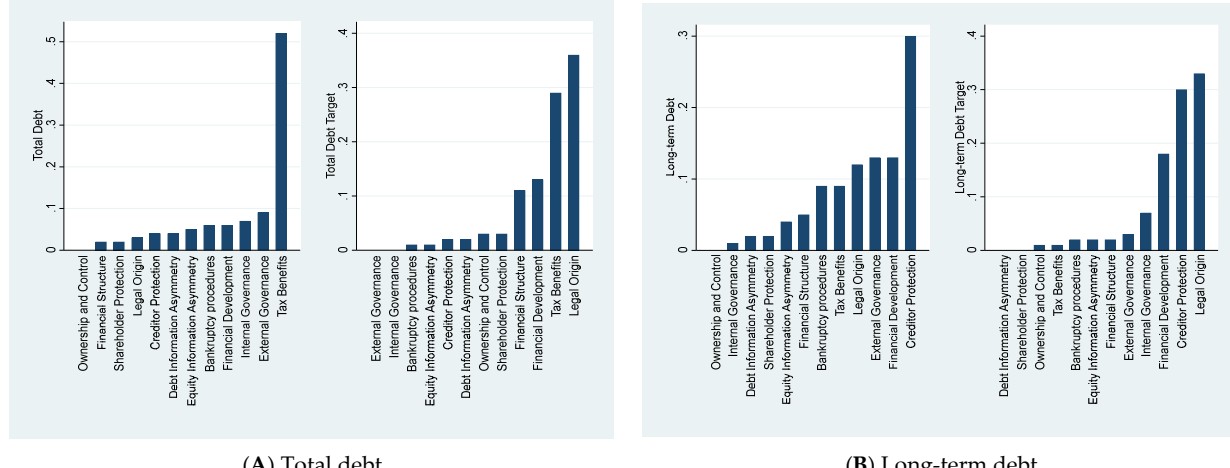

**(A)** Total debt　　　　　　　　　　　　　　　　**(B)** Long-term debt

**Figure 1.** Variance decomposition analyses of actual and target capital structures and the effects of institutions and financial development. Notes: The figure plots the Type III partial sum of squares that is normalized by the sum across the effects of the institutional and financial development determinants estimated from Equation (6). The results for target leverage rely on Target 1, estimated using Blundell and Bond's (1998) two-step system GMM. The definitions and the sources of the variables appear in Table 1.

### 4.2.1. Leverage Targets, Institutions, and the Financial Environment

What are the factors that cause cross-country differences in leverage targets? In Table 5, portfolios were formed to reflect several broad legal and financial traditions and specific institutional dimensions, and *t*-tests of difference in the estimated targets were conducted across the weak and strong institutional portfolios using the total and long-term debt ratios in Panel A and Panel B, respectively.

**Table 5.** The impact of institutions and financial development on target capital structure.

| Supply Channel Portfolios | Legal Origin | | Financial Development | | Financial Structure | | Shareholder Protection | | Internal Governance | | External Governance | |
|---|---|---|---|---|---|---|---|---|---|---|---|---|
| | Civil | Common | Developing | Developed | Bank | Market | Poor | Strong | Poor | Strong | Poor | Strong |
| **Panel A: Total Debt** | | | | | | | | | | | | |
| Panel A1. Target 1 | | | | | | | | | | | | |
| Mean | 26.96 | 24.32 | 28.15 | 23.63 | 26.47 | 23.74 | 24.72 | 24.02 | 26.89 | 23.42 | 27.13 | 23.93 |
| Difference | 2.64 | *** | 4.52 | *** | 2.73 | *** | 0.69 | *** | 3.46 | *** | 3.73 | *** |
| Panel A2. Target 2 | | | | | | | | | | | | |
| Mean | 25.88 | 23.21 | 27.89 | 23.68 | 25.77 | 23.84 | 24.86 | 24.13 | 26.72 | 23.42 | 27.00 | 23.47 |
| Difference | 2.67 | *** | 4.21 | *** | 1.93 | *** | 0.73 | *** | 3.31 | *** | 3.54 | *** |
| Panel A3. Target 3 | | | | | | | | | | | | |
| Mean | 22.29 | 19.74 | 26.60 | 19.90 | 24.41 | 20.05 | 22.66 | 20.35 | 24.26 | 19.54 | 24.56 | 19.65 |
| Difference | 2.55 | *** | 6.70 | *** | 4.36 | *** | 2.31 | *** | 4.71 | *** | 4.91 | *** |
| **Panel B: Long-Term Debt** | | | | | | | | | | | | |
| Panel B1. Target 1 | | | | | | | | | | | | |
| Mean | 11.91 | 16.00 | 16.25 | 14.47 | 15.26 | 14.55 | 13.65 | 14.58 | 13.08 | 15.09 | 12.61 | 14.89 |
| Difference | 4.09 | *** | 1.78 | *** | 0.71 | *** | 0.93 | *** | 2.02 | *** | 2.28 | *** |
| Panel B2. Target 2 | | | | | | | | | | | | |
| Mean | 13.75 | 16.01 | 19.46 | 14.29 | 17.58 | 14.42 | 13.96 | 15.40 | 13.85 | 15.62 | 13.94 | 15.41 |
| Difference | 2.25 | *** | 5.17 | *** | 3.16 | *** | 1.44 | *** | 1.46 | *** | 1.47 | *** |

| Demand Channel Portfolios | Tax Benefits | | Bankruptcy Procedures | | Creditor Protection | | Ownership and Control | | Equity Information Asymmetry | | Debt Information Asymmetry | |
|---|---|---|---|---|---|---|---|---|---|---|---|---|
| | Low | High | Poor | Strong | Poor | Strong | Low | High | Low | High | Low | High |
| **Panel B3. Target 3** (continued) | | | | | | | | | | | | |
| Mean | 9.38 | 11.74 | 14.02 | 10.74 | 12.75 | 10.83 | 10.82 | 10.98 | 9.70 | 11.36 | 9.21 | 11.20 |
| Difference | 2.36 | *** | 3.28 | *** | 1.92 | *** | 0.16 | *** | 1.66 | *** | 1.99 | * |
| **Panel A. Total Debt** | | | | | | | | | | | | |
| **Panel A1. Target 1** | | | | | | | | | | | | |
| Mean | 23.60 | 24.40 | 26.11 | 23.58 | 26.52 | 23.74 | 25.23 | 23.53 | 23.60 | 26.59 | 23.61 | 25.61 |
| Difference | 0.80 | *** | 2.53 | *** | 2.77 | *** | 1.70 | *** | 2.99 | *** | 2.00 | *** |
| **Panel A2. Target 2** | | | | | | | | | | | | |
| Mean | 23.56 | 25.17 | 25.79 | 23.77 | 26.27 | 23.92 | 25.25 | 23.60 | 23.72 | 26.85 | 23.64 | 26.19 |
| Difference | 1.62 | *** | 2.03 | *** | 2.35 | *** | 1.65 | *** | 3.13 | *** | 2.55 | *** |
| **Panel A3. Target 3** | | | | | | | | | | | | |
| Mean | 20.30 | 20.91 | 23.06 | 19.97 | 24.76 | 20.06 | 22.95 | 19.47 | 19.95 | 24.51 | 19.75 | 23.85 |
| Difference | 0.61 | *** | 3.09 | *** | 4.70 | *** | 3.49 | *** | 4.56 | *** | 4.10 | *** |
| **Panel B. Long-Term Debt** | | | | | | | | | | | | |
| **Panel B1. Target 1** | | | | | | | | | | | | |
| Mean | 11.95 | 15.59 | 12.56 | 14.64 | 15.40 | 14.24 | 12.94 | 15.23 | 14.51 | 13.67 | 14.63 | 13.41 |
| Difference | 3.64 | *** | 2.08 | *** | 1.15 | *** | 2.29 | *** | 0.84 | *** | 1.22 | *** |
| **Panel B2. Target 2** | | | | | | | | | | | | |
| Mean | 12.50 | 15.73 | 14.33 | 14.50 | 17.63 | 14.21 | 14.28 | 14.96 | 15.26 | 14.18 | 15.32 | 14.24 |
| Difference | 3.23 | *** | 0.17 | *** | 3.42 | *** | 0.68 | *** | 1.07 | *** | 1.08 | *** |
| **Panel B3. Target 3** | | | | | | | | | | | | |
| Mean | 8.32 | 12.02 | 9.00 | 10.99 | 12.75 | 10.51 | 9.94 | 11.32 | 10.82 | 10.73 | 10.87 | 10.55 |
| Difference | 3.70 | *** | 1.99 | *** | 2.24 | *** | 1.37 | *** | 0.09 | * | 0.33 | ** |

Notes: Countries are allocated into an institution portfolio based on legal origin (civil, common), financial development (developing, developed), financial structure (bank-based, market-based), and the sample median of the remaining institutional indices. Pairwise comparisons of the leverage targets of the selected portfolios were conducted using *t*-tests during the sample period of 1989–2017. Leverage targets were obtained using three distinct methods. Target 1 was estimated using Blundell and Bond's (1998) two-step system GMM. Target 2 was obtained from Fama and French's (2002) cross-sectional leverage regressions, which were estimated annually (Fama and MacBeth 1973). Target 3 denotes the median industry leverage in each country. *, **, and *** indicate significance at the 10%, 5%, and 1% levels. The definitions and the sources of the variables appear in Table 1.

The effects of institutional features on the targeting behavior of firms could occur due to the ability of firms to obtain external capital from investors (i.e., supply side channel) and/or the need and/or willingness of corporations to raise external capital (i.e., demand-side channel). To facilitate the discussion, the economic arguments and the corresponding results are organized in a way that supports different hypotheses along these two dimensions. The institutional factors related to the "supply" explanations are "Legal Origin", "Financial Development", "Financial Structure", "Shareholder Protection", "Internal Governance", "External Governance", whereas the institutional factors related to "demand" explanations are "Tax Benefits", "Bankruptcy Procedures", "Creditor Protection", "Ownership and Control", "Equity information Asymmetry", and "Debt Information Asymmetry".

The first view, which focuses on the supply side of the financial market, hypothesizes that institutional factors affect the willingness of investors to provide capital and take risks. To test this proposition, the estimated targets were tested across the portfolios formed on the basis of the categorizations from the law and finance literature—namely,

civil and common legal origins, financial sector development, and bank- and market-oriented financial systems as well as on some narrower institutional dimensions—namely, shareholder protection and governance mechanisms.

According to the supply channel embraced by the law and finance literature, strong legal and financial systems would have a negative impact on the leverage targets of firms. La Porta et al. (1997, 1998) documented that common law-originated countries provide better investor protection to creditors and shareholders. Similarly, Levine (2002) argues that more sophisticated financial systems alleviate conflicts of interest between financial agents and lead to the better provision of financial services. It is therefore conceivable that the strength of the legal and the financial systems are associated with more favorable financing terms, improving the ability of firms to obtain external capital from investors. Although lower agency costs of debt common to law-originated countries and financially developed countries potentially increase the propensity to lend, it may reduce entrenched willingness of managers to borrow. In contrast, the higher agency costs of equity in civil law-originated countries and financially developing countries should lead to the lower use of outside equity since debt is expected to be used relatively more often than equity when it is easier to expropriate outside equity holders than debt holders (Jensen and Meckling 1976; Jensen 1986). Accordingly, firms in common law-originated and financially developed countries should optimally choose lower target leverage. Consistent with these propositions, in Table 5, Panel A, firms in common law countries desire 2.6–2.7% less debt than firms operating in civil law countries. The economic impact of financial development (4.2–6.7%) significantly exceeds the effects of the legal system.

Furthermore, in countries in which the legal and financial systems are weak, short-term debt should be preferred over long-term debt. Short-term financing helps alleviate the expropriation of creditors by borrowers, limits the period during which opportunistic managers can exploit their creditors, and allows creditors to evaluate the firm's decisions frequently and to change the terms of the financing before significant losses have accumulated (Myers 1977). These results indicate a positive relationship between the efficiency of a country's legal system and the use of long-term debt. Specifically, in Panel B, common law-originated countries target 2–4% higher long-term debt. Surprisingly, developed countries desire lower amounts of long-term debt, indicating that in economies in which both markets and financial intermediaries are strong, the alleviation of conflicts of interests between financial agents primarily involves the choice of financing method, equity over debt, rather than the composition of debt. Unreported tests indicate that the amount of short-term debt is similarly reduced in developed countries.

By the same token, more efficient financial intermediaries as opposed to better access to financial markets would result in higher leverage targets. This conjecture is based on the premise that bank-based financial systems induce lenders to provide credit at more favorable terms, resulting in firms targeting higher leverage. Specifically, banks facilitate the availability of debt financing by providing the closer monitoring and control of firm management. As predicted, differences between bank-based and market-based financial systems lead to systematic differences in target choices. The test results suggest that a bank-based structure leads to a 2–4% higher optimal debt ratio. Furthermore, firms in bank-based economies may have greater access to long-term financing because this form of financing enables intermediaries to use their comparative advantage in monitoring, whereas firms in market-based economies may have better access to equity. Consistent with this proposition, the results illustrate that firms in bank-based economies are supplied with 1–3% more long-term debt.

Another important determinant of the ability of firms to obtain external capital from investors is the severity of contracting problems and the resulting agency costs. Put differently, conflicts of interest between the firm's insiders and outside investors are important determinants of optimal design of securities (Jensen and Meckling 1976). The degree of these agency problems would, to a great extent, depend on the rights attached to

equity securities and their enforcement (shareholder protection) as well as various internal (e.g., constraints on executives and better corporate governance) and external (e.g., government quality as indicated by lower corruption, risk of expropriation and repudiation, strength of law and order, and enforceability of contracts) disciplinary and monitoring mechanisms that limit managerial discretion and facilitate financial contracting. Higher leverage is another efficient mechanism that reduces discretionary funds and subjects managers to the scrutiny of the financial market, effectively curbing the self-serving behavior of managers (Jensen 1986). The results indicate that higher agency costs of equity, as captured by weaker shareholder protection and inferior internal and external governance, lead to higher desired leverage. The economic magnitude is relatively small with shareholder protection (1–2%) but is larger with governance (3–5%), regardless of whether the governance measure is internal or external. Both higher leverage and, in particular, short-term debt can thus be used effectively to control the self-serving interests of managers (Jensen 1986). Although short-term debt is a relevant mechanism to control the agency costs of equity, the economic effect is relatively small. Higher contracting costs and agency costs of equity lead firms to optimally choose up to 2% less long-term debt.

Turning to the second view—which focuses on the demand-side of the financial market, some factors that could affect the willingness and/or the need of the firm to seek external funds entail institutional features that may be relevant for the optimal choice of securities through their impact on tax benefits, financial distress costs, managerial incentive to avoid losing control and/or the need to diversify and share risk with outside investors, and information asymmetry costs.

Debt tax shields play an important role in capital structure choices (Graham 1996). Firms with higher tax rates have stronger incentives to increase their leverage because of the tax deductibility of interest payments. Consistent with this conjecture, a higher effective tax rate is associated with desired leverage that is 1–2% higher. Furthermore, if there are gains from leverage, under an increasing (decreasing) term structure of interest rates, the tax benefits of debt could be maximized with long-term (short-term) debt (Brick and Palmon 1992). Alternatively, a firm lengthens debt maturity as the tax advantage of debt decreases to ensure that the remaining tax advantage of debt, the net of bankruptcy costs, is not less than the amortized flotation costs (Kane et al. 1985). The results provide support for the notion of the maximization of tax benefits, as firms prefer a 3–4% higher amount of long-term debt when subject to higher tax rates.

The impact of financial distress costs on target leverage is not straightforward. On the one hand, higher distress costs could result in lower optimal leverage because of the potential loss in firm value and the managerial fear of losing control. On the other hand, in countries with a higher liquidation risk, managers can maximize firm value by including more debt in their capital structure, thereby minimizing the cash flow to other stakeholders (Crane 2011). The results indicate that higher distress costs captured by poor (less timely, more costly, and less efficient) bankruptcy procedures leads to a 2–3% higher desired leverage. This finding raises the possibility that firm value maximization can be achieved through higher debt when liquidation risk is prominent. Alternatively, debt maturity is a substitute mechanism for controlling financial distress costs. On the one hand, if the costs of financial distress and the costs of rolling over debt are high, firm value is maximized by choosing longer debt maturity. Exposed to the risk of not being able to refund, firms may prefer to lengthen the maturity of their debt to minimize bankruptcy costs arising from inefficient liquidation due to higher default and liquidity risks (Diamond 1991). On the other hand, firms suffering from high costs of financial distress, exposing their managers to the risk of losing control, benefit the most from committing to leverage reductions and thus may have greater incentive to issue short-term debt (Dangl and Zechner 2021). Higher distress costs are associated with up to 2% lower desired long-term debt in support of the conjecture that firms commit to leverage reductions when distress costs are prominent and that the choice of debt composition is an alternative mechanism for controlling financial distress costs. Fan et al. (2012) reach similar conclusions with

an alternative categorization of the bankruptcy process: the existence of an explicit bankruptcy code. On the one hand, explicit bankruptcy codes and poor bankruptcy procedures are associated with higher liquidation risk and higher amounts of actual and target leverage. On the other hand, explicit bankruptcy codes and strong bankruptcy procedures facilitate the reorganization of a firm, leading to more long-term debt and long-term debt targets, respectively (Djankov et al. 2008a).

Financial distress costs could be mitigated by adjusting the properties of the debt contracts, such as the inclusion of covenants, issuance of secured debt through collateralization of tangible assets (Stulz and Johnson 1985), issuance of convertible debt or debt with warrants (Jensen and Meckling 1976), and shortening the maturity of debt (Myers 1977). The creditors would be able to modify and enforce the debt contracts if and only if they are granted the legal rights. However, stronger creditor rights and their enforcement can lead to inefficient liquidation that terminates the continuation option of a firm's business and hurts stockholders. It can also impose a private cost on entrenched managers who maximize their job security rather than firm value. Therefore, while stronger creditor protection may increase target leverage by reducing financial distress costs and by alleviating the moral hazard problems arising from the divergence of interest between shareholders and debtholders, it may also result in a decrease in the target leverage by reducing managerial willingness to borrow to avoid losing control (Acharya et al. 2011). The results indicate that better creditor protection (stronger rights and enforcement) leads to 2–5% lower target leverage, consistent with the proposition that managers desire less debt to prevent the inefficient liquidation of their firm and to secure control. This result raises the possibility that creditor protection and bond covenants are substitutes in reducing financial distress and debt contracting costs (Qi et al. 2011). Alternatively, firms alter the maturity structure of their debt to minimize these costs. Financial theory indicates that, due to the private benefits of control, entrenched managers may prefer to use long-term debt to guarantee their job security by avoiding liquidation (Barnea et al. 1980; Hart and Moore 1998). Consistent with this proposition, poor creditor protection leads to 1–3% higher long-term debt. This finding indicates that managers fearing the loss of their human capital investment on the firm's liquidation target more long-term debt in countries with lower ex ante contractibility.

Ownership and control are also important considerations for the choice of optimal securities. Risk-averse managers would want to decrease the risk associated with their human capital by engaging in risk-reduction activities in their firm, which would likely result in suboptimal investment policies (Myers 1977; Jensen and Meckling 1976). The manifesting underinvestment problem would lead to a lower demand for external capital. Moreover, managers would be more likely to lower their target leverage to stay clear of bankruptcy in order to diversify their employment risk. These adverse managerial incentives arising from holding a less than optimally diversified portfolio would be more prominent in the case of concentrated ownership, which foregoes risk-sharing benefits that are achieved by diffuse ownership (Admati et al. 1994). It may be similarly argued that by holding large shares of equity, stockholders under concentrated ownership may be similarly motivated to diversify firm operations or desire less debt, just as managers do. With regard to debt maturity, a positive association between ownership and control and long-term debt targets is expected, as longer maturity minimizes the risk of liquidation, maximizes managerial job security, and alleviates debt overhang and the resulting underinvestment problems (Barnea et al. 1980; Hart and Moore 1998; Almeida et al. 2009). The findings are consistent with the motives of managers to insulate themselves from the risk of losing control. Specifically, higher concentration in ownership is associated with up to 3.5% less total debt and 2.3% more long-term debt.

Information asymmetry costs are important determinants of the optimal choice of securities. In the presence of asymmetric information (Myers 1984; Myers and Majluf 1984) or managerial optimism (Heaton 2002), the adverse selection costs of issuing risky securities leads to a preference ranking over financing sources, with firms relying on internal funds first, followed by debt and then equity. Accounting standards; the regulation

of equity securities, including mandatory disclosure, liability standards, public enforcement, and insider trading laws; and the presence of public credit registries are some of the mechanisms that facilitate information sharing among financial agents. In weaker institutional settings in which information asymmetry costs are binding, firms should target higher leverage because higher leverage is an effective signaling tool to mitigate information asymmetry problems. The results indicate that higher information asymmetry in equity and debt markets leads to 3–5% and 2–4% higher desired leverage, respectively. Regarding the impact of information costs on debt composition, because the pricing of long-term debt is more sensitive to changes in firm value, to minimize information asymmetry costs, firms may prefer to issue more short-term debt (Diamond 1991; Fama 1985). Higher information asymmetry costs lead firms to choose less long-term debt.

Additional analyses were conducted to examine whether the results are sensitive to differing ownership structures, target adjustment asymmetry, and target estimation techniques. First, separate subsample analyses were conducted for widely held firms, family firms, and state firms. The results indicate that the effects documented for the full sample do not concentrate among any of the ownership groups that were considered. Second, the results may be contaminated by pooling all international firms and by including country-fixed effects when estimating Target 1 and Target 2 since calculated targets would have absorbed the country effects. To better evaluate the impacts of country-level variables, targets were also calculated within each country and put into portfolios based on institutional characteristics. In addition, to take into account the possibility for asymmetric adjustment for under- and over-leveraged firms, tests were conducted separately for under- and over-levered subsamples. Previous findings persist in both subsamples of under-levered and over-levered firms, regardless of how the target estimation model is specified.

Overall, the target capital structure of a firm reflects the environment and traditions in which it operates. The cross-sectional tests of the fitted targets reveal that estimated target capital structures vary meaningfully and significantly across weak and strong institutional environments, reflecting the various costs and benefits of financing imposed on firms, which is consistent with **Hypothesis 2** (**H2**).

4.2.2. Institutional Influences on Targets during the Global Financial Crisis

Thus far, the documented results indicate that an important determinant of capital structure targeting behavior is a firm's institutional environment, which is consistent with financial theory. However, financial theory also highlights the importance of financial conditions as a viable factor behind a firm's willingness and ability to raise capital. To date, no prior research has investigated the relationship between a country's institutions and a firm's financing choices during the recent global financial crisis. This research represents the first attempt to study the targeting behavior of firms during the 2007–2008 period.

First, poor market conditions should negatively affect a firm's overall demand for and access to capital by raising external financing costs, as reflected in higher distress costs, contracting costs, moral hazard problems, agency costs of debt and equity, and information asymmetry costs. Consistent with this conjecture, the actual capital structures documented in Table 3 indicate significantly lower absolute levels of total and long-term debt during the global financial crisis of 2007–2008 compared to the pre-crisis period of 1989–2006. The relative economic difference is up to 7% for total debt and 15% for long-term debt.

Second, because equity contracts are potentially more susceptible to the aforementioned costs, a financial crisis can also affect the structure of the securities offered, shifting them toward shorter maturities. Consistent with this proposition, depending on the target estimation methodology, firms targeted up to 10% more total debt and 13% less long-term debt in their capital structures during the crisis period than during the pre-crisis period of 1989–2006.

Third, the adverse effects of market conditions on the financing patterns of firms can be moderated in countries with strong legal and financial institutions that subject firms to lower external financing costs. In an extreme situation where investors flee to safer international investments, raising capital may be procyclical in strong institutional settings and may be countercyclical in weaker institutional environments. In either case, the differences between weak and strong institutional settings should become more pronounced during difficult economic times. Table 6 documents the differences in the estimated targets across weak and strong institutional portfolios using total and long-term debt ratios for the recent global financial crisis.

**Table 6.** The impact of institutions on target capital structure during the global financial crisis.

| Supply Channel Portfolios | Legal Origin | | Financial Development | | Financial Structure | | Shareholder Protection | | Internal Governance | | External Governance | |
|---|---|---|---|---|---|---|---|---|---|---|---|---|
| | Civil | Common | Developing | Developed | Bank | Market | Poor | Strong | Poor | Strong | Poor | Strong |
| **Panel A: Total Debt** | | | | | | | | | | | | |
| Panel A1. Target 1 | | | | | | | | | | | | |
| Mean | 30.84 | 26.74 | 32.56 | 25.75 | 31.07 | 26.04 | 28.60 | 26.31 | 30.49 | 25.24 | 29.96 | 25.38 |
| Difference | 4.10 | ***/*** | 6.80 | ***/*** | 5.02 | ***/*** | 2.30 | ***/*** | 5.25 | ***/*** | 4.58 | ***/*** |
| Panel A2. Target 2 | | | | | | | | | | | | |
| Mean | 25.34 | 22.86 | 27.64 | 23.09 | 25.11 | 23.47 | 24.09 | 23.89 | 26.31 | 22.65 | 26.42 | 22.70 |
| Difference | 2.48 | ***/*** | 4.54 | ***/*** | 1.64 | ***/*** | 0.20 | ***/*** | 3.66 | ***/*** | 3.72 | ***/*** |
| Panel A3. Target 3 | | | | | | | | | | | | |
| Mean | 24.55 | 19.88 | 27.12 | 19.63 | 24.34 | 20.09 | 22.61 | 20.49 | 24.51 | 18.99 | 24.58 | 19.15 |
| Difference | 4.67 | ***/*** | 7.50 | ***/*** | 4.25 | ***/*** | 2.11 | ***/*** | 5.52 | ***/*** | 5.43 | ***/*** |
| **Panel B: Long-Term Debt** | | | | | | | | | | | | |
| Panel B1. Target 1 | | | | | | | | | | | | |
| Mean | 11.52 | 14.71 | 16.82 | 12.88 | 15.09 | 13.11 | 13.66 | 13.03 | 13.37 | 13.66 | 12.55 | 13.48 |
| Difference | 3.16 | ***/*** | 3.93 | ***/*** | 2.36 | ***/*** | 0.63 | ***/*** | 0.29 | ***/*** | 0.93 | ***/*** |
| Panel B2. Target 2 | | | | | | | | | | | | |
| Mean | 16.10 | 13.17 | 19.40 | 12.82 | 16.90 | 13.24 | 14.82 | 13.38 | 13.63 | 13.70 | 13.47 | 13.65 |
| Difference | 2.93 | ***/*** | 6.58 | ***/*** | 3.67 | ***/*** | 1.44 | ***/*** | 0.07 | ***/*** | 0.18 | */*** |
| Panel B3. Target 3 | | | | | | | | | | | | |
| Mean | 10.69 | 11.62 | 14.16 | 9.31 | 12.22 | 9.62 | 10.58 | 9.58 | 9.95 | 10.11 | 9.30 | 9.93 |
| Difference | 0.93 | ***/*** | 4.84 | ***/*** | 2.59 | ***/*** | 1.00 | ***/*** | 0.16 | ***/*** | 0.63 | ***/*** |

| Demand Channel Portfolios | Tax Benefits | | Bankruptcy Procedures | | Creditor Protection | | Ownership and Control | | Equity Information Asymmetry | | Debt Information Asymmetry | |
|---|---|---|---|---|---|---|---|---|---|---|---|---|
| | Low | High | Poor | Strong | Poor | Strong | Low | High | Low | High | Low | High |
| **Panel A: Total Debt** | | | | | | | | | | | | |
| Panel A1. Target 1 | | | | | | | | | | | | |
| Mean | 26.35 | 27.07 | 29.33 | 25.62 | 30.58 | 25.88 | 28.59 | 25.14 | 25.85 | 30.48 | 25.52 | 29.75 |
| Difference | 0.72 | ***/*** | 3.71 | ***/*** | 4.70 | ***/*** | 3.44 | ***/*** | 4.62 | ***/*** | 4.23 | ***/*** |
| Panel A2. Target 2 | | | | | | | | | | | | |
| Mean | 23.12 | 24.70 | 25.15 | 23.22 | 25.80 | 23.48 | 25.06 | 22.90 | 23.31 | 26.26 | 23.07 | 25.84 |
| Difference | 1.58 | ***/*** | 1.93 | ***/*** | 2.32 | ***/*** | 2.15 | ***/*** | 2.95 | ***/*** | 2.77 | ***/*** |
| Panel A3. Target 3 | | | | | | | | | | | | |
| Mean | 20.20 | 21.34 | 22.88 | 19.72 | 25.04 | 19.90 | 23.00 | 18.97 | 19.93 | 24.46 | 19.45 | 24.18 |
| Difference | 1.14 | ***/*** | 3.16 | ***/*** | 5.15 | ***/*** | 4.03 | ***/*** | 4.53 | ***/*** | 4.73 | ***/*** |
| **Panel B:** | | | | | | | | | | | | |

| Long-Term Debt | | | | | | | | | | | |
|---|---|---|---|---|---|---|---|---|---|---|---|
| Panel B1. Target 1 | | | | | | | | | | | |
| Mean | 10.95 | 14.58 | 12.35 | 12.97 | 15.77 | 12.51 | 12.59 | 13.77 | 13.04 | 13.43 | 12.87 | 13.78 |
| Difference | 3.63 | ***/*** | 0.62 | ***/*** | 3.26 | ***/*** | 1.18 | ***/*** | 0.38 | ***/*** | 0.91 | ***/*** |
| Panel B2. Target 2 | | | | | | | | | | | |
| Mean | 11.35 | 15.16 | 13.59 | 13.07 | 17.40 | 12.76 | 13.15 | 14.22 | 13.23 | 15.34 | 12.87 | 15.61 |
| Difference | 3.81 | ***/*** | 0.52 | ***/*** | 4.65 | ***/*** | 1.07 | ***/*** | 2.10 | ***/*** | 2.74 | ***/*** |
| Panel B3. Target 3 | | | | | | | | | | | |
| Mean | 7.64 | 11.13 | 8.72 | 9.56 | 12.77 | 9.03 | 9.40 | 10.18 | 9.61 | 10.25 | 9.38 | 10.66 |
| Difference | 3.49 | ***/*** | 0.84 | ***/*** | 3.74 | ***/*** | 0.78 | ***/*** | 0.64 | ***/*** | 1.28 | ***/*** |

Notes: The table notes are identical to those in Table 5 except for the sample period, which reflects the global financial crisis of 2007–2008. The first difference refers to the pairwise comparisons of the leverage targets of the selected portfolios conducted using *t*-tests during the global financial crisis of 2007–2008. The second difference refers to the statistical significance of the difference between the normal times and crisis times of the pairwise comparisons of the leverage targets of the selected portfolios. *, **, and *** indicate significance at the 10%, 5%, and 1% levels. The definitions and the sources of the variables appear in Table 1.

Consistent with **Hypothesis 3** (**H3**), the differences across weak and strong institutional portfolios are generally wider during the financial crisis than during the normal times. Table 6 reports statistical significance tests of the difference between the normal times and crisis times of the pairwise comparisons of the leverage targets of the selected portfolios. Based on the Bonferroni multiple comparison tests (Simes 1986), all differences are significant at the 99% confidence level. The results are also consistent using the Scheffé (1953) and Sidak (1967) multiple comparison tests, both of which are less conservative than the Bonferroni multiple comparison test. However, some caution is warranted in reaching conclusions about the impact of the global financial crisis, as there are mixed results for the financial structure, shareholder protection, taxes, and equity information asymmetry using the total debt ratio as well as for the legal traditions, internal and external governance, tax benefits, bankruptcy procedures, and ownership and control using the long-term debt ratio. With the exception of internal governance and bankruptcy procedures, the variance decomposition analyses indicate that these institutional features have lower explanatory power for the leverage targets relative to the observed leverage during the time of the crisis. Specifically, during the crisis, the explanatory power of financial structure, shareholder protection, and equity information asymmetry for the leverage target relative to the observed leverage (as measured by the normalized Type III partial sum of squares for the leverage target divided by the corresponding value for the observed leverage) each drops to one-third of their level at normal times using the total debt ratio. Similarly, using the long-term debt ratio, the explanatory power of legal origin, external governance, taxes, ownership, and control on the leverage target relative to the observed leverage each dropped by 26%, 86%, 76%, and 18%, respectively, during the crisis compared to normal times.

4.2.3. Institutional Influences on Targets Using the Composition of Debt

So far, the analysis of targeting behavior has relied on the absolute level of long-term borrowing and not the mix of long-term and short-term debt. In this subsection, analyses were conducted on the ratio of long-term debt to total debt rather than the level of long-term debt relative to the total assets. Table 7 documents the differences in the estimated targets across weak and strong institutional portfolios using the ratio of long-term debt to total debt.

**Table 7.** The impact of institutions on target capital structure using the composition of debt.

| Supply Channel Portfolios | Legal Origin | | Financial Development | | Financial Structure | | Shareholder Protection | | Internal Governance | | External Governance | |
|---|---|---|---|---|---|---|---|---|---|---|---|---|
| | Civil | Common | Developing | Developed | Bank | Market | Poor | Strong | Poor | Strong | Poor | Strong |
| **Panel A. Target 1** | | | | | | | | | | | | |
| Mean | 46.10 | 62.89 | 54.70 | 57.44 | 55.09 | 57.43 | 53.05 | 56.82 | 46.02 | 60.39 | 43.34 | 59.51 |
| Difference | 16.80 | *** | 2.74 | *** | 2.34 | *** | 3.78 | *** | 14.37 | *** | 16.17 | *** |
| **Panel B. Target 2** | | | | | | | | | | | | |
| Mean | 46.90 | 62.90 | 55.44 | 57.60 | 53.57 | 57.81 | 52.54 | 57.21 | 45.28 | 60.85 | 43.28 | 59.66 |
| Difference | 16.34 | *** | 2.16 | *** | 4.24 | *** | 4.68 | *** | 15.57 | *** | 16.38 | *** |
| **Panel C. Target 3** | | | | | | | | | | | | |
| Mean | 46.74 | 70.74 | 56.65 | 63.57 | 57.20 | 63.57 | 55.63 | 62.83 | 45.09 | 67.65 | 41.79 | 66.43 |
| Difference | 23.81 | *** | 6.92 | *** | 6.37 | *** | 7.20 | *** | 22.56 | *** | 24.64 | *** |
| **Demand Channel Portfolios** | **Tax Benefits** | | **Bankruptcy Procedures** | | **Creditor Protection** | | **Ownership and Control** | | **Equity Information Asymmetry** | | **Debt Information Asymmetry** | |
| | Low | High | Poor | Strong | Poor | Strong | Low | High | Low | High | Low | High |
| **Panel A. Target 1** | | | | | | | | | | | | |
| Mean | 48.20 | 59.90 | 46.33 | 58.02 | 55.29 | 56.23 | 48.74 | 60.48 | 57.57 | 48.55 | 58.17 | 49.16 |
| Difference | 11.69 | *** | 11.68 | *** | 0.94 | *** | 11.73 | *** | 9.02 | *** | 9.02 | *** |
| **Panel B. Target 2** | | | | | | | | | | | | |
| Mean | 48.26 | 60.24 | 45.74 | 58.36 | 55.82 | 56.42 | 48.49 | 60.98 | 57.86 | 47.98 | 58.64 | 48.10 |
| Difference | 11.98 | *** | 12.61 | *** | 0.60 | *** | 12.49 | *** | 9.88 | *** | 10.54 | *** |
| **Panel C. Target 3** | | | | | | | | | | | | |
| Mean | 50.42 | 66.84 | 45.42 | 64.64 | 58.07 | 62.10 | 49.44 | 68.27 | 63.90 | 48.79 | 64.83 | 49.81 |
| Difference | 16.41 | *** | 19.21 | *** | 4.03 | *** | 18.83 | *** | 15.11 | *** | 15.01 | *** |

Notes: A measure of debt composition is defined as the ratio of long-term debt to total debt. Debt targets were obtained using three distinct methods. Target 1 was estimated using Blundell and Bond's (1998) two-step system GMM. Target 2 was obtained from Fama and French's (2002) cross-sectional leverage regressions, which were estimated annually (Fama and MacBeth 1973). Target 3 denotes the median industry leverage in each country. Countries were allocated into an institution portfolio based on legal origin (civil, common), financial development (developing, developed), financial structure (bank-based, market-based), and the sample median of the remaining institutional indices. Pairwise comparisons of the debt targets of the selected portfolios were conducted using *t*-tests during the sample period of 1989–2017. *** indicates significance at the 1% level. The definitions and the sources of the variables appear in Table 1.

The results employing this alternative measure of debt are consistent with the results on long-term debt reported previously, with few meaningful differences. Consistent with the previously reported results, the proportion of long-term debt in total debt is significantly higher in countries with strong (common law) legal environments (16–24%), stronger shareholder protection (4–7%), better internal (14–23%) and external (16–25%) governance measures, higher tax benefits (12–16%), stronger bankruptcy procedures and higher ownership concentration (12–19%), and lower information asymmetry costs (9–15%), which is similar to the results documented with the ratio of long-term debt to total assets.

However, some discrepancies in the results also arise. While prior results on the level of long-term debt have indicated that firms in developed economies target lower absolute levels, possibly because of a greater emphasis on equity financing, the targeted proportion of long-term debt to total debt is 2–7% higher in these economies. The current results indicate a positive relationship between financial system development and the relative use of long-term debt. In financially underdeveloped economies, short-term financing helps to alleviate the expropriation of creditors by borrowers, limits the period during which opportunistic managers can exploit their creditors, and allows creditors to evaluate a firm's decisions frequently and to change the terms of financing before significant losses

have accumulated. Similarly, firms in market-based economies have better access to long-term debt, possibly because developed stock markets transmit useful information to creditors and make lending to these firms less risky.

Another discrepancy in the results arises from the effect of creditor protection on the composition of debt. Weaker creditor protection induces firms to target higher absolute levels of long-term debt, consistent with the better ability of long-term debt to resolve agency problems driven by private benefits of control, moral hazard, and information asymmetry. Yet, the targeted proportion of long-term debt to total debt is 1–4% lower. This finding is consistent with the better ability of short-term debt to alleviate the conflict between stockholders and bondholders arising from underinvestment incentives (Myers 1977; Fama 1985; Stulz and Johnson 1985). Alternatively, short-term debt reduces the problems of risky asset substitution, as prices of short-term bonds are less sensitive to changes in the risk of the underlying asset (Barnea et al. 1980).

### 4.2.4. Target Deviations, Financial Development, and the Financial Crisis

The theory posits that the deviations from optimal leverage should be short-lived in institutional settings where financial frictions could be ameliorated. In contrast, target deviations could have longer lasting effects on the cpital structures of firms at times where transaction costs are binding, preventing firms (and especially financially constrained firms) from making active adjustments to their leverage.

To test whether the target deviations are in line with these predictions of the theory, the impact of institutional and financial factors on (absolute) leverage deviations were assessed initially. In Table 8, *t*-tests of difference in the estimated target deviations were conducted across weak and strong institutional portfolios using the total and long-term debt ratios in Panels A and Panel B, respectively. Consistent with the dynamic tradeoff theory and Öztekin and Flannery (2012), the results suggest that deviations from optimal leverage are less prevalent in strong institutional settings where financial frictions are less binding.

Next, a logistic model was adopted for the effects of financial development and financial crisis on target deviations. Table 9 presents the estimation results for the three alternative target estimation methods, using total debt to total assets, long-term debt to total assets, and long-term debt to total debt, respectively. Panel A and Panel B document the results for all of the sample firms and the financially constrained firms, respectively. A firm is defined as financially unconstrained if it satisfies any of the following conditions in its country: (1) long-term debt issuances to total assets are greater than the 66th percentile, (2) long-term debt reductions to total assets are greater than the 66th percentile, (3) equity issuances to the total assets are greater than the 66th percentile, (4) equity repurchases to the total assets are greater than the 66th percentile. Firms that do not meet any of the above criteria are treated as financially constrained. The dependent variable is a binary variable that takes a value of one for firms that are in the highest or lowest deviation tercile and zero for firms that are in the middle tercile with respect to their deviation from the target (firms close to their target). Target deviations refer to the distance (absolute value) between the estimated targets and the observed (lagged) capital structure. Standard errors are corrected for heteroskedasticity and clustering at the country level. Regressions include unreported industry and year fixed effects to account for shocks to profitability, investment, and stock prices. To alleviate reverse causality concerns, initial deviation is used as an additional control variable.

**Table 8.** The impact of institutions and financial development on target deviations.

| Supply Channel Portfolios | Legal Origin | | Financial Development | | Financial Structure | | Shareholder Protection | | Internal Governance | | External Governance | |
|---|---|---|---|---|---|---|---|---|---|---|---|---|
| | Civil | Common | Developing | Developed | Bank | Market | Poor | Strong | Poor | Strong | Poor | Strong |
| **Panel A: Total Debt Deviation** | | | | | | | | | | | | |
| Panel A1. Deviation from Target 1 | | | | | | | | | | | | |
| Mean | 12.03 | 9.91 | 11.95 | 11.06 | 11.13 | 11.11 | 11.25 | 10.92 | 12.29 | 10.95 | 12.44 | 10.92 |
| Difference | 2.12 | *** | 0.89 | *** | 0.03 | *** | 0.33 | *** | 1.34 | *** | 1.52 | *** |
| Panel A2. Deviation from Target 2 | | | | | | | | | | | | |
| Mean | 14.31 | 13.40 | 14.04 | 12.68 | 14.16 | 11.41 | 14.35 | 12.05 | 14.07 | 14.02 | 14.11 | 13.96 |
| Difference | 0.91 | *** | 1.36 | *** | 2.76 | *** | 2.30 | *** | 0.05 | *** | 0.15 | *** |
| Panel A3. Deviation from Target 3 | | | | | | | | | | | | |
| Mean | 14.27 | 13.43 | 14.08 | 12.30 | 14.20 | 11.18 | 14.39 | 11.82 | 14.07 | 13.87 | 14.05 | 13.98 |
| Difference | 0.83 | *** | 1.78 | *** | 3.02 | *** | 2.57 | *** | 0.20 | *** | 0.07 | *** |
| **Panel B: Total Long-Term Debt Deviation** | | | | | | | | | | | | |
| Panel B1. Deviation from Target 1 | | | | | | | | | | | | |
| Mean | 7.64 | 5.38 | 7.05 | 6.76 | 6.79 | 6.68 | 6.83 | 6.45 | 6.89 | 6.74 | 6.83 | 6.63 |
| Difference | 2.26 | *** | 0.29 | *** | 0.12 | *** | 0.38 | *** | 0.15 | *** | 0.20 | *** |
| Panel B2. Deviation from Target 2 | | | | | | | | | | | | |
| Mean | 12.08 | 9.56 | 11.50 | 11.12 | 11.23 | 10.31 | 11.38 | 9.82 | 11.40 | 10.69 | 11.25 | 10.58 |
| Difference | 2.52 | *** | 0.37 | *** | 0.93 | *** | 1.56 | *** | 0.71 | *** | 0.67 | *** |
| Panel B3. Deviation from Target 3 | | | | | | | | | | | | |
| Mean | 11.37 | 8.21 | 10.31 | 9.49 | 10.41 | 8.56 | 10.52 | 8.39 | 10.64 | 8.97 | 10.50 | 8.90 |
| Difference | 3.16 | *** | 0.83 | *** | 1.84 | *** | 2.12 | *** | 1.67 | *** | 1.60 | *** |

| Demand Channel Portfolios | Tax Benefits | | Bankruptcy Procedures | | Creditor Protection | | Ownership and Control | | Equity Information Asymmetry | | Debt Information Asymmetry | |
|---|---|---|---|---|---|---|---|---|---|---|---|---|
| | Low | High | Poor | Strong | Poor | Strong | Low | High | Low | High | Low | High |
| **Panel A: Total Debt Deviation** | | | | | | | | | | | | |
| Panel A1. Deviation from Target 1 | | | | | | | | | | | | |
| Mean | 14.24 | 13.37 | 14.00 | 13.95 | 11.76 | 11.12 | 0.76 | 0.48 | 13.23 | 14.04 | 13.10 | 14.12 |
| Difference | 0.87 | *** | 0.05 | *** | 0.64 | *** | 0.28 | *** | 0.81 | *** | 1.02 | *** |
| Panel A2. Deviation from Target 2 | | | | | | | | | | | | |
| Mean | 25.17 | 23.56 | 25.79 | 23.77 | 14.17 | 12.65 | 0.93 | 0.68 | 23.72 | 26.85 | 23.64 | 26.19 |
| Difference | 1.62 | *** | 2.03 | *** | 1.52 | *** | 0.25 | *** | 3.13 | *** | 2.55 | *** |
| Panel A3. Deviation from Target 3 | | | | | | | | | | | | |
| Mean | 14.05 | 13.87 | 14.02 | 13.45 | 14.20 | 12.36 | 3.27 | 1.80 | 12.91 | 14.09 | 12.94 | 14.17 |
| Difference | 0.18 | *** | 0.57 | *** | 1.84 | *** | 1.47 | *** | 1.18 | *** | 1.24 | *** |
| **Panel B: Long-Term Debt Deviation** | | | | | | | | | | | | |

| Panel B1. Deviation from Target 1 | | | | | | | | | | | | |
|---|---|---|---|---|---|---|---|---|---|---|---|
| Mean | 6.88 | 6.52 | 6.77 | 6.65 | 7.16 | 6.71 | 0.21 | 0.07 | 6.57 | 6.75 | 6.40 | 6.84 |
| Difference | 0.36 | *** | 0.12 | *** | 0.46 | *** | 0.14 | *** | 0.19 | *** | 0.36 | *** |
| Panel B2. Deviation from Target 2 | | | | | | | | | | | | |
| Mean | 11.63 | 10.02 | 11.19 | 10.35 | 11.17 | 11.10 | 2.08 | 0.48 | 10.34 | 11.18 | 10.16 | 11.28 |
| Difference | 1.61 | *** | 0.85 | *** | 0.07 | *** | 1.60 | *** | 0.83 | *** | 1.12 | *** |
| Panel B3. Deviation from Target 3 | | | | | | | | | | | | |
| Mean | 11.19 | 8.43 | 10.40 | 8.52 | 10.24 | 9.56 | 3.68 | 2.93 | 8.42 | 10.40 | 8.69 | 10.49 |
| Difference | 2.76 | *** | 1.88 | *** | 0.68 | *** | 0.75 | *** | 1.98 | *** | 1.80 | *** |

Notes: Countries were allocated into an institution portfolio based on legal origin (civil, common), financial development (developing, developed), financial structure (bank-based, market-based), and the sample median of the remaining institutional indices. Pairwise comparisons of the target deviations of the selected portfolios were conducted using *t*-tests during the sample period of 1989–2017. Leverage targets were obtained using three distinct methods. Target 1 was estimated using Blundell and Bond's (1998) two-step system GMM. Target 2 was obtained from Fama and French's (2002) cross-sectional leverage regressions, which were estimated annually (Fama and MacBeth 1973). Target 3 denotes the median industry leverage in each country. Target deviations are computed as the absolute distance between the estimated targets and the actual lagged leverage. *** indicates significance at the 1% level. The definitions and the sources of the variables appear in Table 1.

**Table 9.** Target deviations, financial development, and the financial crisis.

| | Panel A: All Firms | | | | | | | | |
|---|---|---|---|---|---|---|---|---|---|
| | Total Debt to Total Assets | | | Total Long-Term Debt to Total Assets | | | Total Long-Term Debt to Total Debt | | |
| | Target 1 | Target 2 | Target 3 | Target 1 | Target 2 | Target 3 | Target 1 | Target 2 | Target 3 |
| Stock market capitalization | −0.0359 ** | 0.0041 | −0.1236 *** | 0.0172 | −0.0546 *** | −0.1488 *** | −0.0688 *** | −0.1763 *** | −0.0105 |
| | (0.013) | (0.781) | (0.000) | (0.234) | (0.000) | (0.000) | (0.000) | (0.000) | (0.486) |
| Bond market capitalization | −0.0884 *** | −0.0954 *** | −0.0494 *** | −0.0297 ** | −0.2212 *** | −0.2423 *** | −0.0628 *** | −0.0105 | −0.0632 *** |
| | (0.000) | (0.000) | (0.000) | (0.021) | (0.000) | (0.000) | (0.000) | (0.447) | (0.000) |
| Private credit | −0.0391 *** | −0.0448 *** | −0.1539 *** | −0.1370 *** | −0.0920 *** | −0.2854 *** | −0.0631 *** | −0.1994 *** | −0.0295 * |
| | (0.007) | (0.002) | (0.000) | (0.000) | (0.000) | (0.000) | (0.000) | (0.000) | (0.053) |
| Financial crisis | 0.5376 *** | 0.7871 *** | 0.9720 *** | 0.5587 *** | 1.0952 *** | 1.4132 *** | 0.5133 *** | 0.8254 *** | 0.9725 *** |
| | (0.000) | (0.000) | (0.000) | (0.000) | (0.000) | (0.000) | (0.000) | (0.000) | (0.000) |
| Lagged target deviation | 0.4161 *** | −1.0605 *** | −1.6299 *** | 0.0291 | −2.8641 *** | −2.8755 *** | 0.0117 | 0.6471 *** | 0.4914 *** |
| | (0.000) | (0.000) | (0.000) | (0.514) | (0.000) | (0.000) | (0.622) | (0.000) | (0.000) |
| Observations | 152,994 | 152,994 | 152,994 | 152,994 | 152,994 | 152,994 | 136,125 | 136,125 | 139,308 |
| Odd Ratios: Financial crisis | 1.7118 | 2.1970 | 2.6432 | 1.7481 | 1.6707 | 2.2828 | 2.6444 | 2.9897 | 1.6707 |
| | Panel B: Financially Constrained Firms | | | | | | | | |
| | Total Debt to Total Assets | | | Total Long-Term Debt to Total Assets | | | Total Long-Term Debt to Total Debt | | |
| | Target 1 | Target 2 | Target 3 | Target 1 | Target 2 | Target 3 | Target 1 | Target 2 | Target 3 |
| Stock market capitalization | −0.0190 | 0.2003 | −0.1431 *** | 0.0178 | −0.0391 ** | −0.1679 *** | −0.0795 *** | −0.1601 *** | −0.0225 |
| | (0.291) | (0.182) | (0.000) | (0.322) | (0.033) | (0.000) | (0.000) | (0.000) | (0.224) |
| Bond market capitalization | −0.0855 *** | 0.0038 | −0.0560 *** | −0.0485 *** | −0.2277 *** | −0.2502 *** | −0.0656 *** | 0.0010 | −0.0873 *** |
| | (0.000) | (0.988) | (0.000) | (0.001) | (0.000) | (0.000) | (0.000) | (0.950) | (0.000) |
| Private credit | −0.0842 *** | −0.1698 | −0.1527 *** | −0.1488 *** | −0.1170 *** | −0.3463 *** | −0.0593 *** | −0.1322 *** | −0.0663 *** |
| | (0.000) | (0.110) | (0.000) | (0.000) | (0.000) | (0.000) | (0.003) | (0.000) | (0.001) |
| Financial crisis | 0.5942 *** | 0.9759 *** | 1.1267 *** | 0.7526 *** | 1.1425 *** | 1.5761 *** | 0.6873 *** | 0.9881 *** | 1.1763 *** |
| | (0.000) | (0.009) | (0.000) | (0.000) | (0.000) | (0.000) | (0.000) | (0.000) | (0.000) |
| Lagged target deviation | 0.3008 *** | −1.7146 *** | −1.4197 *** | −0.1056 | −2.3908 *** | −3.1301 *** | −0.0020 | 0.4543 *** | 0.3332 *** |

| | (0.000) | (0.000) | (0.000) | (0.169) | (0.000) | (0.000) | (0.951) | (0.000) | (0.000) |
|---|---|---|---|---|---|---|---|---|---|
| Observations | 152,994 | 152,994 | 152,994 | 152,994 | 152,994 | 152,994 | 136,125 | 136,125 | 139,308 |
| Odd Ratios: Financial crisis | 1.8115 | 2.6535 | 3.0855 | 2.1224 | 3.1344 | 4.8361 | 1.5880 | 2.6860 | 3.2424 |

Notes: Target deviation is defined as the distance (absolute value of the difference) between the estimated target and observed (lagged) capital structure, using book leverage, for both the total debt ratio and long-term debt ratio. The dependent variable is a binary variable that takes a value of 1 for firms that are in the highest or lowest deviation tercile and 0 for firms that are in the middle tercile, with respect to their deviation from the target (firms close to their target). Financial crisis dummy takes the value of 1 for the years 2007 and 2008 and takes a value of 0 otherwise. Panel A and Panel B document the results for all of the sample firms and the financially constrained firms, respectively. A firm is defined as financially unconstrained if it satisfies any of the following conditions in its country: (1) long-term debt issuances to total assets are greater than the 66[th] percentile, (2) long-term debt reductions to total assets are greater than the 66th percentile, (3) equity issuances to total assets are greater than the 66th percentile, (4) equity repurchases to total assets are greater than the 66th percentile. On the other hand, firms that do not meet any of the above criteria are treated as financially constrained. Leverage targets were obtained using three distinct methods. Target 1was estimated using Blundell and Bond's (1998) two-step system GMM. Target 2was obtained from Fama and French's (2002) cross-sectional leverage regressions, which were estimated annually (Fama and MacBeth 1973). Target 3 denotes the median industry leverage in each country. The regressions include unreported industry and year fixed effects. P-values of the two-tailed tests for the significance of the coefficients corrected for heteroskedasticity and clustering at the country level appear beneath the coefficient estimates in parentheses. *, **, and *** indicate significance at the 10%, 5%, and 1% levels. The definitions and the sources of the variables appear in Table 1.

Variation in transaction costs of accessing the capital markets as captured by the financial development of the country and the financial crisis times is an important driver of target deviations. In general, all three country-level indicators of stock market development, bond market development, and the availability of private credit are associated with a lower likelihood of target deviations. The odds ratios that capture the effect of a change in the independent variables on the implied likelihood of the target deviations reflect the economic magnitude of these effects. In untabulated tests, the odds of deviating are lower by up to 17 percentage points with higher stock market capitalization, up to 22 percentage points with higher bond market capitalization, and up to 25 percentage points with a higher supply of private credit. The costs of transacting in capital markets are significantly higher in hard times: the likelihood of target deviation was much higher during the recent global financial crisis. This effect is economically very significant, increasing the odds of deviating up to 264 and 299 percentage points (the last row of Panel A) with total and long-term debt, respectively. Furthermore, as can be seen in Panel B, the negative impact of the financial crisis on the likelihood of target deviations is amplified within the subsample of the financially constrained firms. As indicated in the last row of each panel, in seven out of nine instances, the odds of deviating during a financial crisis go up by a significant amount, confirming the causal negative effect of an external supply shock on the target deviations. The only exceptions are observed with the total long-term debt to total debt ratio, using Target 1 and Target 2, where the odd ratios decline from 2.6444 to 1.5880 and 2.9897 to 2.6860 for the financially constrained firms compared to the full sample. Note that in Panel B, the upper bound of the odd ratio is 4.8361 using Target 3 for long-term debt. Thus, the deviation likelihood for a financially constrained firm during the financial crisis is 4.83 times that of normal times. Depending on the leverage measure and the target specification, the financial crisis increased the probability of deviation up to five times for the financially constrained firms.

Additional robustness analyses led to the same conclusions. First, similar results were obtained with alternative definitions of financial constraints, for example, when financially constrained (unconstrained) firms are defined as those firms that are in the bottom (top) tercile of the payout distribution. Second, our results do not depend on the specification of the dependent variable. For example, dividing the target deviation into quartiles rather than terciles yields similar results. It is important to note that target deviation is a generated variable that is constructed using the leverage target estimates and the ac-

tual (observed) leverage values. The use of an indicator (rather than the continuous) variable to capture the target deviation has the advantage of bypassing the measurement error that could potentially be introduced into the deviation computation if the estimation of the leverage targets entails any noise. Third, to take into account the possibility of asymmetric adjustment for under- and over-leveraged firms, the impacts of institutional factors on target deviations were separately assessed for under- and over-levered firm subsamples, respectively. Even though under- and over-leveraged firms may move toward the targets differently, the results are still well aligned with the earlier conclusions for both subsamples.

To conclude, after controlling for the initial shock to deviations by lagged deviation and ongoing shocks to deviations by industry and time fixed effects, financial development indicators and crisis times are positively and negatively associated with target deviations, respectively. While some caution is necessary when interpreting the results, it is important to note that the results are not likely to be purely driven by the fact that firms in less developed financial systems or firms that have been subject to macroeconomic shocks simply happen to be further away from their target, which is either due to their own characteristics or to constraints imposed by their environment, as regressions account for this possibility by controlling for the initial deviation. Similarly, results do not seem to be an artifact of the firms that are more subject subject to more severe or frequent shocks to their investment, profitability, and stock prices, as any ongoing shocks are controlled by industry- and year-fixed effects. Additionally, consistent with the causal effect of a supply shock, the adverse effects of the financial crisis on target deviations are greater for the financially constrained firms.

Overall, the results are supportive of **Hypothesis 4 (H4)** and confirm that target deviations are bigger and more likely in the presence of higher external financing costs.

### 4.2.5. Other Considerations

Some robustness checks are in order. First, the results using market leverage (defined as the summation of long-term debt and short-term debt divided by total assets minus book equity plus market equity) indicate that the main results are robust to an alternative definition of financial leverage. Second, using ex post information to estimate leverage targets may cause an estimation bias. An alternative method for constructing target leverage proxies that only rely on the current and historical information in the target estimations yields similar conclusions. Third, mechanical mean reversion in leverage can lead to biased target estimates. Dropping zero leverage observations and/or extreme leverage observations (i.e., greater than 90% and less than 10%) leads to similar results. Fourth, it is possible that institutions affect the targeting behavior of each firm individually, though the general prediction may be different because of sample composition effects. To control for some of the firm characteristics, such as firm size, tangibility, and profits, two groups were formed for each particular institutional variable based on the median country in the sample; these groups were then further subdivided according to firm characteristics, using the median firm in each institutional setting. The main results continue to hold. Fifth, the use of individual rather than composite indices to test for the institutional effects on the target capital structures reveals meaningful cross-sectional variations in the fitted targets, similar to the reported results. Finally, to accommodate the disproportionate representation of firms from some of the countries (e.g., U.S., U.K., and Japan) in the sample, we conducted two additional tests. First, we ran our analysis without U.S. firms, which constitute a large proportion of the sample. Second, we used weights that were proportional to the number of firms in each country. Our results continued to hold in both of these tests.

## 5. Discussion

Despite the irrelevance result of Modigliani and Miller (1958, 1963) in perfect capital markets, there is broad consensus in the literature that the financing deicions of firms can affect firm value. Leverage decisions are shown to have important implications for the corporate as well as real outcomes for both SMEs and listed firms from both developed and developing economies (e.g., Anton 2019). However, the determinants of these decisions are not well understood. Survey evidence suggests that target leverage is an important consideration for firm managers (Graham and Harvey 2001). However, capital structure theories have different predictions regarding whether firms pursue capital structure targets. The pecking order, market timing, and inertia theories argue that there is no target capital structure. The trade-off theory states that firms actively pursue targets and almost immediately reverse deviations from their target leverage. Similarly, dynamic trade-off theory posits that firms have leverage targets but also considers the market imperfections that cause delays in capital structure adjustments. Systemic testing of the leverage targets and target deviations allows us to better understand which theory of capital structure is supported by the data.

Studies testing the static tradeoff model of capital structure generally regress the observed leverage ratios on the proxies for costs of financial distress and agency conflicts and tax benefits of debt with the idea that the model is a good first order approximation of the equilibrium. Studies testing the capital structure dynamics generally examine a partial adjustment model that assumes some form of target adjustment and examines the extent to which leverage changes in response to profitability shocks, stock price fluctuations, and the financing needs of firms. By showing that the estimates of target leverage and target deviations are meaningful and congruent with the theory and that the target leverage choices of firms vary plausibly with institutional costs and benefits and financial conditions around the globe, our study adds to both genres. Our novel empirical test of leverage targeting behavior thus lends robust support to the dynamic trade-off theory.

Our findings have important implications for investors, scholars, and policymakers. This study suggests that focusing on observed leverage choices may yield biased and misleading inferences with respect to the relative importance of institutional factors for optimal capital structure determination as well as the importance of deviations from the optimal levels. This, in turn, may yield to suboptimal regulatory interventions and policies, possibly leading to inferior corporate outcomes with negative repercussions on firm growth and valuation. Importantly, these biases would be particularly more pronounced during financial distress times when financially constrained firms would benefit greatly from regulatory interventions and public policies. Consequently, the proper modeling of corporate target capital structures is critical to policymakers in developing the necessary instruments to assist firms facing financing and valuation challenges, particularly in weak institutional settings that are more exposed to financial frictions.

## 6. Conclusions

This article examines the leverage targeting behavior of firms in 46 countries during the period of 1989–2017. Financial theory indicates that a firm's debt ratio targets are the outcomes of optimization, such that the firm trades off the costs and benefits of debt and equity, as reflected in various firm, industry, macroeconomic, and country attributes. However, are the estimates of the target capital structures and deviations from the target consistent with the theory? This question has not been directly assessed in previous research.

This paper empirically tests the target behavior of firms by assessing whether the firm actions associated with the targets and target deviations inferred from various leverage models are consistent with the predictions of the theory, as opposed to existing studies that only investigate actual leverage and capital structure adjusting actions by firms. The

approach of examining target behavior is distinct from that of the previous studies focusing on observed leverage determination and is motivated by three important considerations. Firms cannot always promptly react to the deviations from their optimal structure, and consequently, cross-sectional variations in actual capital structures could be observed, even across a sample of firms having the same target capital structures. Similarly, shocks to firms could lead to fluctuations in target debt ratios over time without necessarily causing changes in observed debt ratios. Finally, even though leverage targets are a function of observed leverage ratios, the extent to which the cost and benefit factors are reflected in the observed leverage as opposed to in the leverage targets and/or and the target deviations could vary across firms and over time.

This article estimates target capital structures using reliable empirical approaches. Estimated leverage targets differed significantly from actual leverage regardless of the empirical methodology used to estimate the targets and the sample period. The economic magnitudes of the target deviations relative to the actual capital structures ranged from an average of 2.68–11.83% and 1.19–23.95% with the total and long-term debt ratios, respectively. Deviations widened during the financial crisis compared with the period before the crisis by 93.08% and 14.50% for total and long-term debt ratios, respectively.

The cross-sectional tests of the fitted targets reveal that the estimated target capital structures varied significantly across institutional and financial environments, reflecting the various costs and benefits of financing imposed on firms, which is consistent with the theory. Firms from countries with common law origin, stronger shareholder protection, better governance, superior bankruptcy procedures, more concentrated ownership and control, and lower information asymmetry costs target lower leverage and higher long-term debt, whereas market-based and developed financial systems, lower tax benefits, and stronger creditor protection are associated with lower total and long-term debt targets. These findings provide support for the applicability of the selected estimation methods to appropriately measure leverage targets among the sample firms.

To shed light on the impact of a country's institutional setting on the targeting behavior of firms during the crisis period, capital structures during the global financial crisis of 2007–2008 were compared to the pre-crisis period of 1989–2006. Deteriorating market conditions negatively affect a firm's demand for and access to capital by raising external financing costs. Capital structures indicated significantly lower absolute levels of both total and long-term debt during the global financial crisis. The relative economic difference is up to 7% when using total debt and 15% when using long-term debt. Furthermore, the desired structure of the securities offered shifts toward shorter maturities. Firms targeted up to 10% more total debt and 13% less long-term debt in their capital structures during the crisis period compared to the pre-crisis period of 1989–2006. Furthermore, the adverse effects of market conditions on the financing patterns of firms were less pronounced in countries with strong legal and financial institutions.

Finally, financial development indicators as captured by stock market development, bond market development, and the availability of private credit are all associated with the lower probability of target deviations. The costs of transacting in capital markets are significantly higher in hard times. The likelihood of target deviation was up to three times higher during the recent global financial crisis in the overall sample. However, the negative shock to the supply and to the cost of external finance adversely affected financially constrained firms in particular, increasing their odds of deviation up to five times.

Overall, the empirical results indicate that estimates of target leverage and target deviations are meaningful and congruent with the theory, and the optimal capital structure choices of firms vary plausibly with institutional costs and benefits and financial conditions around the globe. Thus, our proposed empirical test of leverage targeting behavior lends strong support to the dynamic trade-off theory. Yet, an important limitation of this study is that the empirical approach of systematically testing leverage targets and target deviations across institutional settings may not be sufficient alone to arrive at a conclusion on leverage

targeting behavior. As future research, another reliable approach would involve constructing a theoretical model and forming target proxies from this model, which would require running simulations and then constructing a statistical test to compare observed leverage with target leverage while incorporating the distribution of these variables.

**Author Contributions:** Conceptualization, A.G. and Ö.Ö.; methodology, A.G. and Ö.Ö.; software, A.G. and Ö.Ö.; validation, A.G. and Ö.Ö.; formal analysis, A.G. and Ö.Ö.; investigation, A.G. and Ö.Ö.; resources, A.G. and Ö.Ö.; data curation, A.G. and Ö.Ö.; writing—original draft preparation, A.G. and Ö.Ö.; writing—review and editing, A.G. and Ö.Ö.; visualization, A.G. and Ö.Ö.; supervision, A.G. and Ö.Ö.; project administration, A.G. and Ö.Ö. All authors have read and agreed to the published version of the manuscript.

**Funding:** This research received no external funding.

**Institutional Review Board Statement:** Not applicable.

**Informed Consent Statement:** Not applicable.

**Data Availability Statement:** Subscription datasets were analyzed in this study.

**Conflicts of Interest:** The authors declare no conflict of interest.

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
