# Peer review of "Financial Leverage and Debt Maturity Targeting: International Evidence"

_jrfm, doi:10.3390/jrfm14090437_

Round 1

Reviewer 1 Report

This is a very important topic and worth researching, nonetheless the paper lacks scientific rigor currently and needs polishing/developing. In order to make the paper publishable I would ask you to include the following changes:

Major changes:

(1) The structure of each scientific article should be: introduction + theory + hypotheses development + methodology + empirical findings + scientific discussion + Please use the classical structure of the article:

  1. Introduction
  2. Literature review and hypotheses development
  3. Material and methods
  4. Results
  5. Discussion
  6. Conclusion
(2) What is the novelty of your article? It would be good if you provided an even sharper argumentation for your research gap. So to summarize, adding some more conceptual detail to your introduction would do a big favor to your theoretical rigor. In the Introduction section please put more emphasis on the novelty of this paper, what new is brought by this paper to the literature?   (3) There is no ‘Literature Review’ section which is very essential. Theoretical Framework is far too general. What is the state of the art in the existing literature? In the Literature Review section try to make a short summary what are the current research (previous studies) in this field and what is your contribution, how you will continue the extant literature. What research gaps exist? Please identify it.   (4) In this article there is little scientific discussion, and you should compare your results with the results of prior studies and previous empirical investigations done by different authors. Please develop this part of your article.   (5) In the Conclusions section of your article please remember that we need to have four compulsory elements: (i) general summary of empirical results, (iii) implications for practice, (iii) Research Limitations and (iv) suggestions for further research in this topic. I really miss some of them (e.g. limitations). Please complete it.   (6) In my opinion the submission does not present required contribution. Please show what is your contribution?  

Author Response

Authors’ Responses to Reviewer 1

We are very grateful for your constructive suggestions and comments. We have addressed the issues you raised and have incorporated them in the current version of the paper. We believe these changes have substantially improved the paper. Please find below a brief account of our revisions. The text includes a more comprehensive treatment of the changes. Our responses are in italic.

We hope you enjoy reading the revised version of our manuscript and find it ready for publication at the Journal of Risk and Financial Management.

This is a very important topic and worth researching, nonetheless the paper lacks scientific rigor currently and needs polishing/developing.

Thank you for your encouraging comments and constructive feedback. Following your suggestions, we have made improvements to the organization and writing in the revised version of the manuscript. We have also revised the theoretical framework to clearly articulate our hypotheses and to better link them to the empirical results. We believe that these steps have significantly enhanced the scientific rigor of our paper.

In order to make the paper publishable I would ask you to include the following changes:

 (1) The structure of each scientific article should be: introduction + theory + hypotheses development + methodology + empirical findings + scientific discussion + Please use the classical structure of the article:

  1. Introduction
  2. Literature review and hypotheses development
  3. Material and methods
  4. Results
  5. Discussion
  6. Conclusion

We appreciate this suggestion. The current organization of the paper now reflects your suggested structure.  Please note that we replaced “material” with “sample”, since “material” does not directly apply to our research. Also, we followed your suggestion and added two new sections to the paper: “Literature review and hypotheses development” and “Discussion”.

(2) What is the novelty of your article? It would be good if you provided an even sharper argumentation for your research gap. So to summarize, adding some more conceptual detail to your introduction would do a big favor to your theoretical rigor. In the Introduction section please put more emphasis on the novelty of this paper, what new is brought by this paper to the literature?  

Thank you for this suggestion. We have substantially revised the introduction to emphasize the novelty and contribution of the paper. The text now provides an even sharper argumentation for our research gap, but we provide a brief summary here.

Prior research has not directly tested and established whether the estimates of target capital structures are meaningful. This article addresses this research gap by investigating the significance and the determinants of capital structure targets and target deviations and systematically testing the estimated leverage and debt maturity targets and target deviations across different environments. Thus, an important novelty of the paper is a new empirical test of leverage targeting behavior based on estimates of leverage targets and target deviations. Prior studies testing the tradeoff theory of capital structure regress observed leverage ratios on proxies for costs of financial distress and agency conflicts and tax benefits of debt with the idea that the model is a good first order approximation of the equilibrium. This approach is problematic since firms do not typically operate at their optimal leverage due to financial frictions, indicating that observed capital structure is a noisy proxy of target capital structure. In other words, while the current debate about the role of national institutions is focused around the observed leverage and observed debt maturity as a proxy for optimal leverage and optimal debt maturity; this study estimates optimal leverage and optimal debt maturity with reliable empirical models and uses these estimates as its key measures in subsequent tests.

This new empirical approach improves upon the traditional approach of prior studies that examine the cross-sectional determinants of observed leverage ratios for three important reasons. Firstly, if firms cannot instantaneously offset the movements from their optimal capital structure, as posited by the dynamic tradeoff theory, cross-sectional variations in actual capital structures could be observed even across a sample of firms sharing the same target capital structures. Secondly, shocks to firms’ profitability and investment opportunities, stock prices, and other factors may cause variations in target debt ratios over time without necessarily having the same impact on observed debt ratios. Thirdly, the extent to which various cost and benefit factors affect observed leverage ratios as opposed to leverage targets could vary across firms and/or over time. Consequently, leverage models that assume that firms always operate at their long-run equilibrium and proxy for the target capital structure using the observed capital structure, while assuming that target deviations would be statistically and economically trivial fail to properly account for the true firm capital structure dynamics. The results confirm that misleading conclusions may be reached regarding the relative influences of the institutional factors for leverage targeting behavior of corporate firms since observed leverage is a noisy proxy of target leverage.

In addition, while many papers assess the determinants of a firm’s capital structure, most of these studies examine firms within a single country, typically the United States. Studies with an international scope typically ignore the dynamic nature of the data or focus on observed (and not target) capital structure in the cross -sectional tests. For example, Fan, Titman, and Twite (2012) examine how the institutional environment influences observed capital structure choices. Their analysis implicitly assumes instantaneous adjustment to optimal capital structure and mostly emphasizes supply side financing (i.e., investors). While the empirical analysis is dynamic in some papers, it doesn’t assess institutional influences on debt maturity choices and it doesn’t focus on targets and target deviations. In this paper, we study how cross-country variations in institutional variables affect the choices of target leverage and target debt maturity in a dynamic, international setting.

Furthermore, in contrast to some other studies (e.g. Fan, Titman, and Twite (2012), this study employs a large set of institutional features that are not only related to the supply side, but also related to the demand side (i.e., corporations), reflecting various costs (e.g. bankruptcy costs, agency costs, and information asymmetry costs) that firms face in their country.

Several other important features of our study similarly improve upon prior studies. Firstly, target capital structure is estimated using three distinct empirical approaches to ensure that a particular model specification does not drive the results. Secondly, the international sample is the most comprehensive of the capital structure studies conducted to date and spans 46 countries and 29 years. The large cross-section and longer time series not only provide more powerful tests but also yield novel results. Thirdly, the availability of post-2006 data creates a novel environment in which to demonstrate how the global financial crisis has affected target choices and target deviations. The recent global financial crisis had a significant impact on firms around the world, yet little is known about how it affected firms’ capital structure choices and the adjustment to target leverage. This study adds to the existing literature by testing the impact of the recent financial crisis on leverage targets and target deviations for the first time. The current study also analyzes how the recent global financial crisis has affected the relationships among leverage targets and a country’s institutional strength. Finally, the use of three distinct leverage measures – 1) absolute level of total borrowing, 2) absolute level of long-term borrowing, and 3) the mix of long-term debt in total debt – allow for more refined tests of the impact of the institutional, financial, and macroeconomic environment on firms’ capital structure targeting behavior.

In addition to more clearly articulating our contributions to the literature in the introduction, we have added Section 2 “Literature Review and Hypotheses Development”, and Section 5 “Discussion” to further discuss and elaborate on our contributions. It is important to note that, to enhance the theoretical rigor, further discussion of the capital structure theories is undertaken in Section 5 “Discussion”. The implications of our findings for investors, scholars, and policymakers are also articulated in the same section. Moreover, Section 2.2 “Hypotheses Development” elaborates on the theory behind hypotheses development and clearly lists four distinct hypotheses that are being tested. Finally, in the empirical results section, Section 4, the empirical results are more clearly tied to these hypotheses.

(3) There is no ‘Literature Review’ section which is very essential. Theoretical Framework is far too general. What is the state of the art in the existing literature? In the Literature Review section try to make a short summary what are the current research (previous studies) in this field and what is your contribution, how you will continue the extant literature. What research gaps exist? Please identify it.  

Following your suggestion, Section 2 presents “Literature Review and Hypotheses Development” section. Subsection 2.1 presents “Literature Review”.  Based on your recommendation, this section sets the theoretical framework, makes a short summary of the current and previous research in the field, identifies the research gaps, and articulates the contribution of this paper to the field.

(4) In this article there is little scientific discussion, and you should compare your results with the results of prior studies and previous empirical investigations done by different authors. Please develop this part of your article.  

Thank you for this suggestion. The revised manuscript provides a comparison of our results with those of prior studies. As an example, on page 21, we compare our findings to those of Fan, Titman, and Twite (2012): “Fan, Titman, and Twite (2012) reach similar conclusions with an alternative categorization of the bankruptcy process: the existence of an explicit bankruptcy code. On the one hand, explicit bankruptcy codes and poor bankruptcy procedures are associated with higher liquidation risk and consequently higher amounts of actual and target leverage. On the other hand, explicit bankruptcy codes and strong bankruptcy procedures facilitate the reorganization of a firm, leading to more long-term debt and long-term debt targets, respectively. (Djankov et al. 2008).”

(5) In the Conclusions section of your article please remember that we need to have four compulsory elements: (i) general summary of empirical results, (iii) implications for practice, (iii) Research Limitations and (iv) suggestions for further research in this topic. I really miss some of them (e.g. limitations). Please complete it.  

Thank you for your constructive suggestion. We now articulate the limitation of our research as well as suggestion for future research in the conclusion as follows: “Overall, the empirical results indicate that estimates of target leverage and target deviations are meaningful and congruent with the theory, and firms’ choices of optimal capital structure vary plausibly with institutional costs and benefits and financial conditions around the globe. Thus, our proposed empirical test of leverage targeting behavior lends strong support to the dynamic trade-off theory. Yet, an important limitation of this study is that the empirical approach of systematically testing leverage targets and target deviations across institutional settings may not be sufficient alone to arrive at a conclusion on leverage targeting behavior. As future research, another reliable approach would involve constructing a theoretical model and forming target proxies from this model, which would require running simulations and then constructing a statistical test to compare observed leverage with target leverage while incorporating the distribution of these variables.”

(6) In my opinion the submission does not present required contribution. Please show what is your contribution?  

As explained in more detail in our response to your comment #2, we hope that the revised discussion in the introduction, the newly added Section 2 “Literature Review and Hypotheses Development”, and the newly added Section 5 “Discussion” now collectively present a better account of the paper’s contributions to the literature.

Reviewer 2 Report

The paper provides evidence on leverage and debt maturity targeting in a large international setting. The key findings are: Targets and target deviations are plausibly influenced by the institutional environment. Firms from countries with strong legal institutions target lower leverage and higher long-term debt, whereas better-functioning financial systems result in lower target leverage and long-term debt. Financial crisis has shifted the desired structure of the securities toward shorter maturities and led to more prevalent target deviations. The topic is interesting and within the scope of the journal.

Main comments

1.The paper is well structured and written.

2.The title of the article is clear and adequate.

3.The abstract is clear, it presents the object of research, the content, and the results.

4.The introduction states the objectives of the paper and the relevance of the research work.

  1. The methodology seems sound.
  2. The results and interpretations are correct.

Major concerns

In the introduction, the authors should stress the importance of leverage for both SMEs and listed firms, both from developed and developing economies (see recommended readings).

Implications

You should clarify the contributions of the paper which are not elaborated well in the current paper. You can talk about the following contributions: What insights can you provide based on your finding? Do they push forward our understanding? What should we do with your research? Do you have any suggestions to improve the current regulation or practice? Adding the above discussion may help you make more contributions and position your contributions better.   

The authors should highlight the limits of their research. The last section of the conclusion should summarize all your findings, their implications to researchers and practitioners, the future direction for research, the limitations of the current study, etc.

 Minor concern

The paper is not formatted using the JRFM format.

In conclusion, I would like to thank the authors for a very interesting, unique, and potentially important paper. Hope these comments and suggestions can help further their study. I consider that the paper can bring a significant contribution to the extant literature once the above-mentioned recommendations are taken into account.

Suggested readings:

  1. Leverage and firm growth: an empirical investigation of gazelles from emerging Europe, International Entrepreneurship and Management Journal, 15(1), 209–232, doi: 10.1007/s11365-018-0524-5.

Author Response

Authors’ Responses to Reviewer 2

We are very grateful for your constructive suggestions and comments. We have addressed the issues you raised and have incorporated them in the current version of the paper. We believe these changes have substantially improved the paper. Please find below a brief account of our revisions. The text includes a more comprehensive treatment of the changes. Our responses are in italic.

The paper provides evidence on leverage and debt maturity targeting in a large international setting. The key findings are: Targets and target deviations are plausibly influenced by the institutional environment. Firms from countries with strong legal institutions target lower leverage and higher long-term debt, whereas better-functioning financial systems result in lower target leverage and long-term debt. Financial crisis has shifted the desired structure of the securities toward shorter maturities and led to more prevalent target deviations. The topic is interesting and within the scope of the journal.

Main comments

1.The paper is well structured and written.

2.The title of the article is clear and adequate.

3.The abstract is clear, it presents the object of research, the content, and the results.

4.The introduction states the objectives of the paper and the relevance of the research work.

  1. The methodology seems sound.
  2. The results and interpretations are correct.

Thank you for your encouraging comments and constructive feedback. Following your suggestions, we have made improvements to the organization and writing in the revised version of the manuscript.

In the introduction, the authors should stress the importance of leverage for both SMEs and listed firms, both from developed and developing economies (see recommended readings).

Thank you for this helpful suggestion. The introduction now emphasizes the importance of capital structure for both SMEs and listed firms, from developed as well as developing economies as our first point. We also articulate this point again in the discussion of our findings in Section 5.  The following paper is now added to our reference list: Anton, Sorin Gabriel, 2019. Leverage and firm growth: an empirical investigation of gazelles from emerging Europe. International Entrepreneurship and Management Journal 15: 209–232.

You should clarify the contributions of the paper which are not elaborated well in the current paper. You can talk about the following contributions: What insights can you provide based on your finding? Do they push forward our understanding? What should we do with your research? Do you have any suggestions to improve the current regulation or practice? Adding the above discussion may help you make more contributions and position your contributions better.   

Thank you for this suggestion. We have substantially revised the manuscript to emphasize the novelty and contribution of the paper. The text now provides more details on the insights our research provides based on our findings and implications for current regulation and practice, but we provide a brief summary here.

Prior research has not directly tested and established whether the estimates of target capital structures are meaningful. This article addresses this research gap by investigating the significance and the determinants of capital structure targets and target deviations and systematically testing the estimated leverage and debt maturity targets and target deviations across different environments. Thus, an important novelty of the paper is a new empirical test of leverage targeting behavior based on estimates of leverage targets and target deviations. Prior studies testing the tradeoff theory of capital structure regress observed leverage ratios on proxies for costs of financial distress and agency conflicts and tax benefits of debt with the idea that the model is a good first order approximation of the equilibrium. This approach is problematic since firms do not typically operate at their optimal leverage due to financial frictions, indicating that observed capital structure is a noisy proxy of target capital structure. In other words, while the current debate about the role of national institutions is focused around the observed leverage and observed debt maturity as a proxy for optimal leverage and optimal debt maturity; this study estimates optimal leverage and optimal debt maturity with reliable empirical models and uses these estimates as its key measures in subsequent tests.

This new empirical approach improves upon the traditional approach of prior studies that examine the cross-sectional determinants of observed leverage ratios for three important reasons. Firstly, if firms cannot instantaneously offset the movements from their optimal capital structure, as posited by the dynamic tradeoff theory, cross-sectional variations in actual capital structures could be observed even across a sample of firms sharing the same target capital structures. Secondly, shocks to firms’ profitability and investment opportunities, stock prices, and other factors may cause variations in target debt ratios over time without necessarily having the same impact on observed debt ratios. Thirdly, the extent to which various cost and benefit factors affect observed leverage ratios as opposed to leverage targets could vary across firms and/or over time. Consequently, leverage models that assume that firms always operate at their long-run equilibrium and proxy for the target capital structure using the observed capital structure, while assuming that target deviations would be statistically and economically trivial fail to properly account for the true firm capital structure dynamics. The results confirm that misleading conclusions may be reached regarding the relative influences of the institutional factors for leverage targeting behavior of corporate firms since observed leverage is a noisy proxy of target leverage.

In addition, while many papers assess the determinants of a firm’s capital structure, most of these studies examine firms within a single country, typically the United States. Studies with an international scope typically ignore the dynamic nature of the data or focus on observed (and not target) capital structure in the cross -sectional tests. For example, Fan, Titman, and Twite (2012) examine how the institutional environment influences observed capital structure choices. Their analysis implicitly assumes instantaneous adjustment to optimal capital structure and mostly emphasizes supply side financing (i.e., investors). While the empirical analysis is dynamic in some papers, it doesn’t assess institutional influences on debt maturity choices and it doesn’t focus on targets and target deviations. In this paper, we study how cross-country variations in institutional variables affect the choices of target leverage and target debt maturity in a dynamic, international setting.

Furthermore, in contrast to some other studies (e.g. Fan, Titman, and Twite (2012), this study employs a large set of institutional features that are not only related to the supply side, but also related to the demand side (i.e., corporations), reflecting various costs (e.g. bankruptcy costs, agency costs, and information asymmetry costs) that firms face in their country.

Several other important features of our study similarly improve upon prior studies. Firstly, target capital structure is estimated using three distinct empirical approaches to ensure that a particular model specification does not drive the results. Secondly, the international sample is the most comprehensive of the capital structure studies conducted to date and spans 46 countries and 29 years. The large cross-section and longer time series not only provide more powerful tests but also yield novel results. Thirdly, the availability of post-2006 data creates a novel environment in which to demonstrate how the global financial crisis has affected target choices and target deviations. The recent global financial crisis had a significant impact on firms around the world, yet little is known about how it affected firms’ capital structure choices and the adjustment to target leverage. This study adds to the existing literature by testing the impact of the recent financial crisis on leverage targets and target deviations for the first time. The current study also analyzes how the recent global financial crisis has affected the relationships among leverage targets and a country’s institutional strength. Finally, the use of three distinct leverage measures – 1) absolute level of total borrowing, 2) absolute level of long-term borrowing, and 3) the mix of long-term debt in total debt – allow for more refined tests of the impact of the institutional, financial, and macroeconomic environment on firms’ capital structure targeting behavior.

In addition to more clearly articulating our contributions to the literature in the introduction, we have added Section 2 “Literature Review and Hypotheses Development”, and Section 5 “Discussion” to further discuss and elaborate on our contributions. It is important to note that, to enhance the theoretical rigor, further discussion of the capital structure theories is undertaken in Section 5 “Discussion”. The implications of our findings for investors, scholars, and policymakers are also articulated in the same section. Moreover, Section 2.2 “Hypotheses Development” elaborates on the theory behind hypotheses development and clearly lists four distinct hypotheses that are being tested. Finally, in the empirical results section, Section 4, the empirical results are more clearly tied to these hypotheses.

We hope that the revised discussion in the introduction, the newly added Section 2 “Literature Review and Hypotheses Development”, and the newly added Section 5 “Discussion” now collectively present a better account of the paper’s contributions to the literature.

The authors should highlight the limits of their research. The last section of the conclusion should summarize all your findings, their implications to researchers and practitioners, the future direction for research, the limitations of the current study, etc.

Thank you for your constructive suggestion. We now articulate the limitation of our research as well as suggestion for future research in the conclusion as follows: “Overall, the empirical results indicate that estimates of target leverage and target deviations are meaningful and congruent with the theory, and firms’ choices of optimal capital structure vary plausibly with institutional costs and benefits and financial conditions around the globe. Thus, our proposed empirical test of leverage targeting behavior lends strong support to the dynamic trade-off theory. Yet, an important limitation of this study is that the empirical approach of systematically testing leverage targets and target deviations across institutional settings may not be sufficient alone to arrive at a conclusion on leverage targeting behavior. As future research, another reliable approach would involve constructing a theoretical model and forming target proxies from this model, which would require running simulations and then constructing a statistical test to compare observed leverage with target leverage while incorporating the distribution of these variables.”

The paper is not formatted using the JRFM format.

Thank you for pointing this out. We have applied some formatting changes, such as moving the footnotes to the main text and reformatting the references to comply with the journal formatting requirements. If there are any other pending formatting issues, the paper will be formatted again using the JRFM format upon acceptance.

In conclusion, I would like to thank the authors for a very interesting, unique, and potentially important paper. Hope these comments and suggestions can help further their study. I consider that the paper can bring a significant contribution to the extant literature once the above-mentioned recommendations are taken into account.

Thank you for your insightful comments and suggestions. We hope you enjoy reading the revised version of our manuscript and find it ready for publication at the Journal of Risk and Financial Management.

Suggested readings:

  1. Leverage and firm growth: an empirical investigation of gazelles from emerging Europe, International Entrepreneurship and Management Journal, 15(1), 209–232, doi: 10.1007/s11365-018-0524-5.
